# Null-Space Filtering for Data-Free Continual Model Merging: Preserving Stability, Promoting Plasticity

**Zihuan Qiu[1,3], Lei Wang[1], Yang Cao[3], Runtong Zhang[1], Bing Su[3], Yi Xu[2],
Fanman Meng[1]\*, Linfeng Xu[1], Qingbo Wu[1], Hongliang Li[1]**
[1]University of Electronic Science and Technology of China, Chengdu, China
[2]Dalian University of Technology, Dalian, China  [3]Jiigan Technology

## Abstract

Data-free continual model merging (DFCMM) aims to fuse independently fine-tuned models into a single backbone that evolves with incoming tasks without accessing task data. This paper revisits two fundamental desiderata for DFCMM: *stability*, avoiding interference with earlier tasks, and *plasticity*, adapting faithfully to each new task. This poses a challenge that existing approaches fail to address: how to bridge data-level desiderata with parameter-space optimization to ensure stability and plasticity in the absence of task data. To this end, we propose NUFILT (**NU**ll-space **FILT**ering), a data-free framework that directly links these desiderata into parameter-space optimization. Our key observation is that task vectors approximately align with representation subspaces, providing structural surrogates for enforcing stability and plasticity. Accordingly, we design a null-space projector that preserves prior responses by filtering overlapping components of new task vectors, ensuring stability. We further introduce a lightweight LoRA adapter that injects complementary task-specific signals to enable plasticity. The adapter is trained with a projection-based surrogate loss that preserves consistency with prior knowledge while introducing novel directions. This joint filtering–adaptation process enables the backbone to absorb new knowledge while retaining existing behaviors, with updates fused back in a layer-wise linear fashion without extra parameters or inference cost. Theoretically, we establish approximate subspace alignment guarantees that justify null-space filtering. Empirically, NUFILT achieves state-of-the-art performance with minimal forgetting on both vision and NLP benchmarks, improving average accuracy by 4–7% over OPCM and WUDI-Merging, while narrowing the gap to fine-tuning and reducing computation overhead. The code is available at: `https://github.com/zihuanqiu/NUFILT`

## 1 Introduction

Modern machine learning systems are often deployed in dynamic environments where tasks arrive one after another. Training a new model from scratch for each task is computationally expensive and requires retaining all historical data, which is often impractical. Maintaining a separate checkpoint for every task also becomes infeasible due to memory and deployment constraints (McMahan et al., 2017; Fang et al., 2024; Qiu et al., 2024). A more appealing alternative is to reuse existing models—either drawn from a model library or trained for new tasks—and consolidate their knowledge into a single backbone that evolves over time (Zhou et al., 2024b; Yu et al., 2024b; Huang et al., 2024a; Tang et al., 2025). However, this consolidation is expected to be performed while retaining only the merged model for storage efficiency, and without accessing any data in order to safeguard privacy, which makes the problem particularly challenging. These constraints highlight the urgent need for approaches that integrate knowledge across sequential tasks directly in parameter space without extra storage or data costs.

---

*Corresponding author

Recent studies investigate data-free continual model merging (DFCMM) (Liu & Soatto, 2023; Porrello et al., 2025; Tang et al., 2025), where tasks arrive sequentially and, at each step, only the newly fine-tuned task model and the previously merged backbone are available (see Fig. 1). We revisit two fundamental desiderata for DFCMM: *stability*, avoiding interference with earlier tasks, and *plasticity*, adapting faithfully to each new task. While these desiderata are inherently defined at the data level, existing approaches lack a principled translation into parameter-space objectives. As a result, simple averaging or arithmetic updates often cause interference that undermines stability (Izmailov et al., 2018; Ilharco et al., 2023); projection methods fail to preserve plasticity when task vectors are correlated (Tang et al., 2025; Yadav et al., 2023); and adaptive strategies typically rely on auxiliary data, which violates the data-free constraint (Yang et al.; Tang et al., 2024b; Qiu et al., 2025). Achieving both stability and plasticity under the strict requirements of DFCMM remains an open challenge.

In this paper, we introduce NUFILT (**NU**ll-space **FILT**ering), a novel framework for DFCMM that directly bridges data-level desiderata with parameter-space optimization. The key observation is that task vectors exhibit approximate alignment with representation subspaces, enabling structural surrogates for enforcing stability and plasticity without data. To achieve stability, a null-space projector filters out components of the new task vector overlapping with earlier subspaces, suppressing interfering activations. To achieve plasticity, a lightweight LoRA adapter performs projection-aware adaptation, optimized with a data-free surrogate loss: updates are constrained to remain consistent with previous tasks while introducing complementary signals along directions unique to the new task. Finally, the projector, task vector, and adapter are fused back into the backbone in a layer-wise linear fashion, so up-

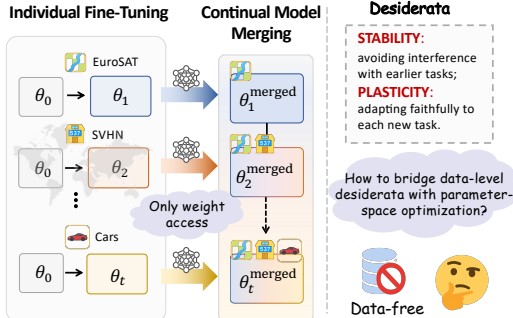

Figure 1: Illustration of **data-free continual model merging** (DFCMM). At each step, only the current task model and the previously merged model are accessible, and the merging process is performed without access to any data. The merged model is expected to preserve prior knowledge (stability) while adapting efficiently to new tasks (plasticity).

dates are absorbed without extra parameters or inference cost. Theoretically, we provide guarantees of approximate subspace alignment, establishing a rigorous foundation for null-space filtering. Empirically, we demonstrate that NUFILT achieves state-of-the-art accuracy with minimal forgetting across both vision and NLP tasks. Compared to recent methods such as OPCM (Tang et al., 2025) and WUDI-Merging (Cheng et al., 2025), NUFILT improves average accuracy by 4–7%, while substantially narrowing the gap to individual fine-tuning and reducing computation overhead.

To summarize, our main contributions are as follows:

- We establish that task vectors exhibit approximate alignment with data representation subspaces, revealing a geometric property that explains how task vectors interact with representations.
- We propose NUFILT, a data-free continual model merging framework that combines null-space filtering with projection-aware adaptation to enforce both stability and plasticity.
- Extensive experiments on vision and NLP benchmarks demonstrate that NUFILT achieves state-of-the-art performance, surpassing prior methods in accuracy while better resisting forgetting.

## 2 RELATED WORK

**Model Merging.** Model merging provides an efficient alternative to multi-task or continual training by combining multiple fine-tuned models directly in parameter space, without revisiting their training data. Early efforts adopted simple weight averaging (Utans, 1996; Shoemake, 1985), which can connect models through linear mode connectivity (Entezari et al., 2021; Ainsworth et al., 2022) but often suffers from severe performance degradation due to parameter interference. Task Arithmetic (TA) (Ilharco et al., 2023) formalizes merging through task vectors, making merging operations more explicit. However, since standard fine-tuning rarely guarantees disentangled updates (Ortiz-Jimenez et al., 2023), TA can amplify interference, prompting structured finetuning strategies (Ortiz-Jimenez

et al., 2023; Liu & Soatto, 2023; Porrello et al., 2025). To mitigate conflicts, several methods introduce additional structure. Sparsity-based approaches such as TIES-Merging (Yadav et al., 2023) prune redundant or conflicting parameters, while reweighting schemes (Yu et al., 2024b) balance task contributions. Projection-based methods enforce orthogonality between task vectors to separate task directions (Tang et al., 2025; Cheng et al., 2025), but struggled when tasks are inherently correlated. Adaptive strategies further adjust merging process with auxiliary calibration data (Yang et al.; Tang et al., 2025; Qiu et al., 2025), but this departs from the strictly data-free setting. Overall, these methods reduce interference to varying degrees but either rely on data or assume strong independence between tasks, leaving the core challenge of simultaneously achieving stability and plasticity in the data-free continual setting largely unresolved.

**Continual Learning.** Continual learning tackles catastrophic forgetting (McCloskey & Cohen, 1989), where models lose performance on earlier tasks when adapting to new ones. Solutions include constraining parameter updates via importance weights (Kirkpatrick et al., 2016; Zenke et al., 2017; Aljundi et al., 2018), preserving knowledge through distillation (Hou et al., 2019; Douillard et al., 2020), or replaying exemplars and surrogates (Rebuffi et al., 2016; Liu et al., 2021). Other strategies expand capacity with dynamic architectures (Lee et al., 2017; Qiu et al., 2023; Zhou et al., 2024a) or adopt parameter-efficient modules such as adapters and prompts (Yu et al., 2024a; Huang et al., 2024b). Beyond traditional data-driven approaches, a complementary direction explores continual learning through model merging. Early efforts merged fine-tuned checkpoints to alleviate forgetting (Mirzadeh et al., 2021; Wen et al., 2023; Marczak et al., 2024), but these typically relied on training data or sequential fine-tuning. More recently, continual model merging (Jin et al., 2023; Liu & Soatto, 2023; Porrello et al., 2025; Tang et al., 2025; Qiu et al., 2025) has emerged, aiming to fuse independently fine-tuned models directly in parameter space, avoiding both data access and checkpoint storage while improving scalability and privacy.

**Orthogonality in Task Fusion.** Orthogonality-based methods aim to reduce interference between tasks by projecting updates into orthogonal spaces. These can be categorized into two types: orthogonality between data and parameters (Farajtabar et al., 2020; Wang et al., 2021; Liang & Li, 2024), and orthogonality directly between parameter updates(Wei et al.; Cheng et al., 2025; Tang et al., 2025). Fang et al. (2025) project perturbations onto the null space of preserved knowledge, while Xiong et al. (2024) enforce orthogonality between task vectors, and Wang et al. (2023) learn new parameters within orthogonal low-rank subspaces to reduce interference. In contrast, our method projects each new task vector into the null space of previously merged task representations, but without requiring any task data. This differs from Fang et al. (2025); Wang et al. (2023), which rely on external data to estimate orthogonality constraints. Moreover, unlike Xiong et al. (2024), which requires simultaneous access to all task vectors, our approach operates sequentially using only the previously merged task vector and the current one, enabling efficient continual model merging.

## 3 BACKGROUND AND MOTIVATION

We formalize data-free continual model merging in Sec. 3.1, review representative approaches in Sec. 3.2, outline key desiderata for data-free merging in Sec. 3.3, and analyze the geometric relation between task vectors and representations in Sec. 3.4.

### 3.1 PROBLEM SETTING

We study data-free continual model merging, where a sequence of task-specific models $\{\theta_t\}_{t=1}^T$ are independently fine-tuned from a shared pre-trained model $\theta_0$ on labeled datasets $\mathcal{D}_t$ with disjoint label sets $\mathcal{C}_t$. The goal is to obtain a single merged model $\theta_T^{\text{merged}}$ that generalizes to the union label space $\mathcal{C}_{1:T} = \bigcup_{t=1}^T \mathcal{C}_t$. Unlike conventional continual learning, we assume *no access* to raw training data. All knowledge integration must therefore occur directly in parameter space, via recursive merging of task-adapted checkpoints:

$$\theta_t^{\text{merged}} = \text{Merge}\left(\theta_{t-1}^{\text{merged}}, \theta_t\right), \quad (\theta_1^{\text{merged}} = \theta_1). \tag{1}$$

To expose the underlying structure, let $\tau_t = \theta_t - \theta_0$ denote the *task vector* (Ilharco et al., 2023), capturing the update induced by task $t$. Rather than merging two checkpoints directly, we reinterpret

continual merging as learning a transformed update:

$$\theta_t^{\text{merged}} = \theta_{t-1}^{\text{merged}} + \tilde{\tau}_t, \quad (\tilde{\tau}_t = F(\tau_t)). \tag{2}$$

Here, $F$ transforms task vectors to stay compatible with previously merged parameters, ensuring seamless integration of new updates.

## 3.2 REPRESENTATIVE CONTINUAL MODEL MERGING SOLUTIONS

Several continual model merging approaches can be viewed as special cases of task vector transformation, each with different assumptions on $F(\tau_t)$. Below we outline representative solutions:

❶ **Weight Averaging (WA)** (Izmailov et al., 2018): Updates parameters by simple averaging, $\theta_t^{\text{merged}} = \frac{1}{t}\big[(t-1)\theta_{t-1}^{\text{merged}} + \theta_t\big]$. This stabilizes optimization but assumes task compatibility and equal importance of checkpoints, making it sensitive to semantic conflicts. ❷ **Task Arithmetic (TA)** (Ilharco et al., 2023): Adds scaled task vectors, $\theta_t^{\text{merged}} = \theta_{t-1}^{\text{merged}} + \lambda \tau_t$, where $\lambda$ is a tunable coefficient. It can outperform naive averaging but lacks structural constraints, leading to scale sensitivity and task interference. ❸ **Orthogonal Projection-based Continual Merging (OPCM)** (Tang et al., 2025): Projects each task vector onto the orthogonal complement of previous directions, $\theta_t^{\text{merged}} = \theta_0 + \frac{1}{\lambda_t}\big[\lambda_{t-1}\tau_{t-1}^{\text{merged}} + \mathcal{P}^{(t-1)}(\tau_t)\big]$, where $\mathcal{P}^{(t-1)}(\cdot)$ removes overlapping subspaces. This enforces geometric separation but struggles when task vectors are inherently entangled or non-orthogonal. ❹ **AdaMerging** (Yang et al.): Adapts coefficients using a small unlabeled test set, $\theta_t^{\text{merged}} = \theta_{t-1}^{\text{merged}} + \lambda_t(\mathcal{D}_t^{\text{test}}) \tau_t$. By exploiting test-time signals, it improves task alignment and mitigates scaling issues, but requires auxiliary data, violating the data-free assumption.

## 3.3 DESIDERATA FOR DATA-FREE CONTINUAL MERGING

In continual model merging, updates for a new task should be incorporated without disrupting knowledge accumulated from earlier ones. This reflects the classical stability–plasticity balance in continual learning. To describe how a transformed update $\tilde{\tau}_t^{(l)} = F(\tau_t^{(l)})$ influences the merged model, we use two layer-wise quantities as surrogates for these effects. Let $\theta_0^{(l)} \in \mathbb{R}^{d_o \times d_i}$ be the pretrained weights at layer $l$, $\tau_i^{(l)}$ the task vector for task $i$, and $x^{(l)} \in \mathbb{R}^{d_i}$ the corresponding layer activations.

① **Stability surrogate.** This term measures how much the new update alters the model's responses on earlier tasks. For activations $x^{(l)} \sim \mathcal{D}_{i \leq t-1}$, we define:

$$\mathcal{L}_{\text{stab}}^{(l)}(\tilde{\tau}_t^{(l)}) = \mathbb{E}_{x^{(l)} \sim \mathcal{D}_{i \leq t-1}}\big[\ell((\theta_{t-1}^{\text{merged},(l)} + \tilde{\tau}_t^{(l)})x^{(l)}, \theta_{t-1}^{\text{merged},(l)}x^{(l)})\big]. \tag{3}$$

where $\ell$ is a distance metric (*e.g.*, squared $\ell_2$).

② **Plasticity surrogate.** This term evaluates how well the merged parameters follow the behavior of the task's individual model. For activations $x^{(l)} \sim \mathcal{D}_t$, we define:

$$\mathcal{L}_{\text{plas}}^{(l)}(\tilde{\tau}_t^{(l)}) = \mathbb{E}_{x^{(l)} \sim \mathcal{D}_t}\big[\ell((\theta_{t-1}^{\text{merged},(l)} + \tilde{\tau}_t^{(l)})x^{(l)}, (\theta_0^{(l)} + \tau_t^{(l)})x^{(l)})\big]. \tag{4}$$

These two terms anchor the stability–plasticity trade-off within the parameter space used for merging. However, optimizing $\mathcal{L}_{\text{stab}}^{(l)}(\tilde{\tau}_t^{(l)})$ and $\mathcal{L}_{\text{plas}}^{(l)}(\tilde{\tau}_t^{(l)})$ is not possible in the data-free setting, as neither $\mathcal{D}_{i \leq t-1}$ nor $\mathcal{D}_t$ is accessible. This leads to the central challenge:

> *How can continual merging achieve **stability** and **plasticity** without data?*

## 3.4 APPROXIMATE ALIGNMENT BETWEEN REPRESENTATION AND TASK VECTOR SUBSPACES

We examine the geometric relation between task vectors and data representations. We hypothesize that task vectors tend to align with principal directions in the representation space that capture task-relevant information. To validate this, we conduct a subspace alignment analysis and provide a theoretical guarantee that *task vector subspaces are approximately aligned with data subspaces*, which in turn motivates our null-space filtering approach.

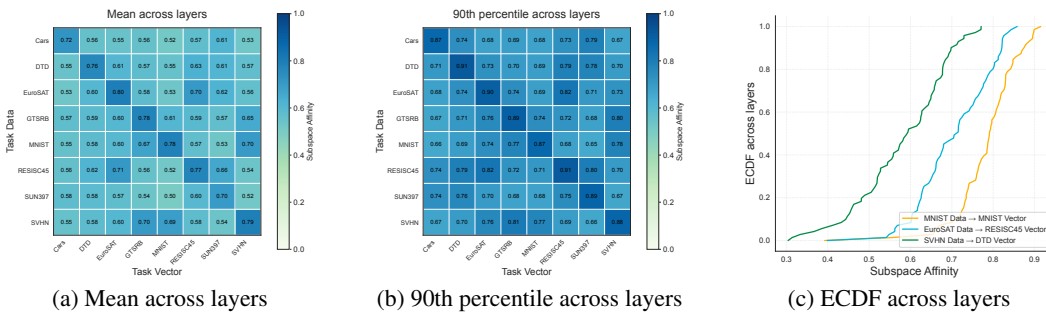

(a) Mean across layers      (b) 90th percentile across layers      (c) ECDF across layers

Figure 2: Subspace affinity between data and task vectors in ViT-B/16 across eight datasets. Heatmaps show diagonal dominance, and layer-wise empirical cumulative distribution functions (ECDFs) confirm higher affinities for matched pairs.

For each task and layer $l$, let $V_d^{(l)} \in \mathbb{R}^{d_i \times r_d}$ be the singular vectors of the representation covariance $\mathbb{E}[x^{(l)}(x^{(l)})^\top]$. For the task vector $\tau^{(l)}$, we compute its truncated SVD, $\tau^{(l)} \approx \hat{U}^{(l)}\hat{\Sigma}^{(l)}\hat{V}^{(l)\top}$, where $\hat{V}^{(l)} \in \mathbb{R}^{d_i \times r_v}$ is the top $r_v$ right singular vectors. We define the subspace affinity:

$$\mathcal{A}(V_d^{(l)}, \hat{V}^{(l)}) = \frac{1}{r_d}\|\hat{V}^\top V_d\|_F^2, \quad (r_d \leq r_v \leq d_i) \tag{5}$$

which lies in $[0, 1]$ and measures the degree of alignment between the data and vector subspace.

Empirically, across ViT-B/16 models fine-tuned on eight datasets, affinity heatmaps (Fig. 2) show strong diagonal dominance: matched data–vector pairs exhibit much higher affinities than mismatched ones. Layer-wise ECDFs (Fig. 2c) further reveal a spectrum of overlaps, from high (MNIST–MNIST), moderate (EuroSAT–RESISC45), to low (SVHN–DTD). These results indicate that task vectors are consistently aligned with task-relevant representation directions, a trend observed in both vision and NLP tasks (additional results for more models provided in the Appendix C.2).

**Theorem 1** (Approximate Subspace Alignment). *Let $\tau^{(l)} \in \mathbb{R}^{d_o \times d_i}$ be the task vector at layer $l$, and $H \in \mathbb{R}^{N \times d_i}$ a representation matrix of rank $r_d$ with right singular vectors $V_d \in \mathbb{R}^{d_i \times r_d}$. Suppose $\tau^{(l)}$ admits the decomposition $\tau^{(l)} = T_0 + E$ with $T_0 = BH$, $\mathrm{rank}(T_0) = r_d$, where each row $E_k$ of $E$ satisfies $\|E_k\|_2 \leq \Psi_k$. Let $\sigma_{r_d}(T_0)$ denote the $r_d$-th largest singular value of the matrix $T_0$. If $\sigma_{r_d}(T_0) > \sqrt{d_o}\max_k \Psi_k$, then for any $r_v \geq r_d$, the span of the top $r_v$ right singular vectors $\hat{V}$ of $\tau^{(l)}$ is approximately aligned with $\mathrm{span}(V_d)$ in the sense that*

$$1 - \mathcal{A}(V_d^{(l)}, \hat{V}^{(l)}) \leq \zeta^2, \tag{6}$$

*where $\mathcal{A}(V_d^{(l)}, \hat{V}^{(l)}) = \frac{1}{r_d}\|\hat{V}^\top V_d\|_F^2$ and $\zeta = \frac{\sqrt{d_o}\max_k \Psi_k}{\sigma_{r_d}(T_0) - \sqrt{d_o}\max_k \Psi_k}$.*

This theorem shows that $\tau^{(l)}$ consists of a low-rank component aligned with the representation space plus a bounded perturbation, so its principal subspace is approximately aligned with the data subspace. Together with the empirical evidence, this provides both geometric intuition and theoretical justification for treating task vectors as carriers of task-relevant representation directions.

## 4 METHOD: NUFILT

Motivated by the above, we propose NUFILT (**NU**ll-space **FILT**ering), a data-free method that enforces structural constraints on $\tau_t$. The key idea is to leverage the geometry of prior tasks to suppress activations tied to earlier ones, enabling continual merging while mitigating forgetting.

### 4.1 STEPS IN NUFILT: FILTERING, ADAPTING & FUSING

The procedure of NUFILT unfolds in three steps, as illustrated in Fig. 3:

**Step ❶: Filtering.** To prevent interference with prior tasks, we introduce a null-space filter before applying each new task vector. For layer $l$, we compute the cumulative update

$$\tilde{\tau}_{\leq t-1}^{(l)} = \theta_{t-1}^{\mathrm{merged},(l)} - \theta_0^{(l)}, \tag{7}$$

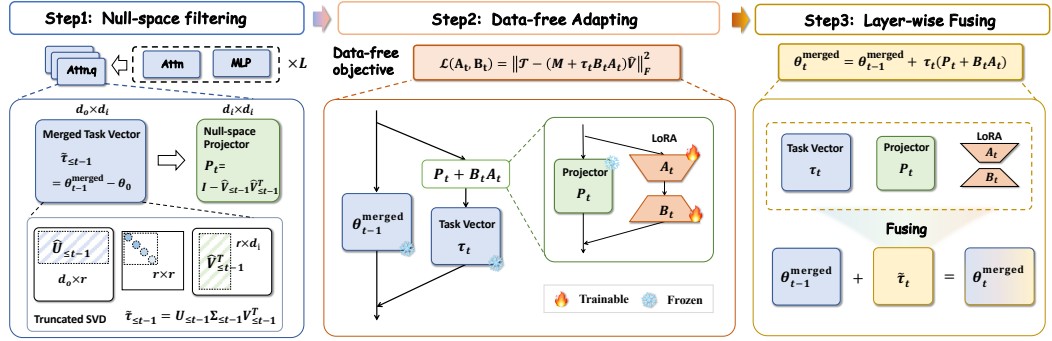

Figure 3: Overview of the NUFILT procedure. ❶ **Filtering**: the new task vector is processed through a null-space projector that suppresses activations from previous tasks, ensuring stability to past knowledge. ❷ **Adapting**: within the filter, a lightweight LoRA adapter refines the update for the current task using a data-free objective. ❸ **Fusing**: the filter, task vector, and LoRA module are merged back into the backbone, keeping the parameter count and inference cost unchanged.

and obtain its top-$r_p$ right singular vectors $\hat{V}^{(l)}_{\leq t-1}$. The null-space filter is defined as

$$P_t^{(l)} = I - \hat{V}^{(l)}_{\leq t-1}\hat{V}^{(l)\top}_{\leq t-1}, \tag{8}$$

which removes components aligned with earlier task directions. Thus, for old task features $x^{(l)} \in \text{span}(\hat{V}^{(l)}_{\leq t-1})$, we have $P_t^{(l)}x^{(l)} = 0$, ensuring stability to prior knowledge.

**Step ❷: Adapting.** Within the filtered subspace, we insert a low-rank LoRA adapter

$$P_t^{(l)} \;\to\; P_t^{(l)} + B_t^{(l)}A_t^{(l)}, \tag{9}$$

where $A_t^{(l)} \in \mathbb{R}^{r_l \times d_i}$ and $B_t^{(l)} \in \mathbb{R}^{d_o \times r_l}$. This lightweight term restores task-specific flexibility and is trained with a data-free objective (Sec. 4.2), allowing the merged model to recover the performance of the fine-tuned model on task $t$ without requiring raw data.

**Step ❸: Fusing.** Finally, the filter, task vector, and LoRA module are merged into the backbone:

$$\theta_t^{\text{merged},(l)} = \theta_{t-1}^{\text{merged},(l)} + \tau_t^{(l)}\big(P_t^{(l)} + B_t^{(l)}A_t^{(l)}\big). \tag{10}$$

Since all operations are linear, the composite update can be absorbed into the weight matrix, leaving the parameter count and inference cost identical to an individual model.

## 4.2 DATA-FREE OBJECTIVE OF NUFILT

From Theorem 1, we obtain a data-free upper bound on parameter–representation interactions.

**Corollary 1** (Data-free upper bound). *Let $X \in \mathbb{R}^{N \times d_i}$ with largest singular value $\sigma_1(X)$, rank $r_d$, and right singular vectors $V_d$. Let $\tau \in \mathbb{R}^{d_o \times d_i}$ with top-$r_v$ right singular vectors $\hat{V}$ ($r_v \geq r_d$). If Theorem 1 ensures $1 - \frac{1}{r_d}\|\hat{V}^\top V_d\|_F^2 \leq \zeta^2$, then for any $\rho \in \mathbb{R}^{d_o \times d_i}$,*

$$\big\|(\rho - \tau)X^\top\big\|_F^2 \;\leq\; 2\,\sigma_1(X)^2\Big(\big\|(\rho - \tau)\hat{V}\big\|_F^2 + r_d\,\zeta^2\,\|\rho - \tau\|_2^2\Big). \tag{11}$$

**Subspace surrogate for losses.** Under squared $\ell_2$ loss, the data losses (Eq. 3–4) are

$$\mathcal{L}_{\text{stab}} = \mathbb{E}_{X \sim \mathcal{D}_{i \leq t-1}}\|(\tilde{\tau}_{\leq t} - \tilde{\tau}_{\leq t-1})X^\top\|_F^2, \quad \mathcal{L}_{\text{plas}} = \mathbb{E}_{X \sim \mathcal{D}_t}\|(\tilde{\tau}_{\leq t} - \tau_t)X^\top\|_F^2, \tag{12}$$

where $\tilde{\tau}_{\leq t} = \sum_{i=1}^t \tilde{\tau}_i$. Let $X_o$ and $X_n$ denote feature matrices from old and new tasks. Applying Corollary 1 with $(\rho, \tau) = (\tilde{\tau}_{\leq t}, \tilde{\tau}_{\leq t-1})$ and $(\tilde{\tau}_{\leq t}, \tau_t)$, and denoting the corresponding misalignments by $\zeta_o$ and $\zeta_n$, we obtain the following *data-free upper bounds*:

$$\|(\tilde{\tau}_{\leq t} - \tilde{\tau}_{\leq t-1})X_o^\top\|_F^2 \leq 2\,\sigma_1(X_o)^2\Big(\|(\tilde{\tau}_{\leq t} - \tilde{\tau}_{\leq t-1})\hat{V}_{\leq t-1}\|_F^2 + r_o\zeta_o^2\|(\tilde{\tau}_{\leq t} - \tilde{\tau}_{\leq t-1})\|_2^2\Big), \tag{13}$$

$$\|(\tilde{\tau}_{\leq t} - \tau_t)X_n^\top\|_F^2 \leq 2\,\sigma_1(X_n)^2\Big(\|(\tilde{\tau}_{\leq t} - \tau_t)\hat{V}_t\|_F^2 + r_n\zeta_n^2\|(\tilde{\tau}_{\leq t} - \tau_t)\|_2^2\Big). \tag{14}$$

---

**Algorithm 1** NUFILT Procedure

---

**Input:** pre-trained model $\theta_0$; fine-tuned models $\{\theta_t\}_{t=1}^T$; rank parameters $\{r_p, r_l, r_v\}$; learning rate $\eta$; and maximum number of iterations MaxIter.

**Initialize:** $\theta_1^{\text{merged}} = \theta_1$

**for** each task $t \in \{2, \ldots, T\}$ **do**

    **for** selected linear layer index $l \in \{1, \ldots, L\}$ **do**

        $P_t^{(l)} = I - \hat{V}_{\leq t-1}^{(l)} \hat{V}_{\leq t-1}^{(l)\top}$            $\triangleright$ null-space via $\hat{V}_{\leq t-1}^{(l)} = \text{TOPRIGHTSVD}(\tilde{\tau}_{\leq t-1}^{(l)}, r_p)$

        $A_t^{(l)} \sim \mathcal{N}(0, \sigma_{init}^2), \quad B_t^{(l)} \leftarrow \mathbf{0}$         $\triangleright$ initialize LoRA components with low rank $r_l$

        **for** iter $= 1$ **to** MaxIter **do**

            compute $M = \tilde{\tau}_{\leq t-1}^{(l)} + \tau_t^{(l)} P_t^{(l)}, \mathcal{T}$, and $\hat{V}$ via Eq. 17

            $\mathcal{L}(A_t^{(l)}, B_t^{(l)}) = \left\| \mathcal{T} - \left( M + \tau_t^{(l)} B_t^{(l)} A_t^{(l)} \right) \hat{V} \right\|_F^2$     $\triangleright$ data-free objective (Eq. 16)

            $A_t^{(l)} \leftarrow A_t^{(l)} - \eta \nabla_{A_t^{(l)}} \mathcal{L}, \quad B_t^{(l)} \leftarrow B_t^{(l)} - \eta \nabla_{B_t^{(l)}} \mathcal{L}$     $\triangleright$ update LoRA components

    $\left\{ \theta_t^{\text{merged},(l)} \right\}_{l=1}^L = \left\{ \theta_{t-1}^{\text{merged},(l)} + \tau_t^{(l)} \left( P_t^{(l)} + B_t^{(l)} A_t^{(l)} \right) \right\}_{l=1}^L$     $\triangleright$ layer-wise fusion

**Output:** $\theta_T^{\text{merged}}$

---

Thus when the misalignment $\zeta$ is small, both losses are governed by the terms $\|(\tilde{\tau}_{\leq t} - \tilde{\tau}_{\leq t-1})\hat{V}_{\leq t-1}\|_F^2$ and $\|(\tilde{\tau}_{\leq t} - \tau_t)\hat{V}_t\|_F^2$, offering a projection-aware objective for effective data-free surrogates.

**Projection-aware objective.** At layer $l$, the merged task vector (subtracting $\theta_0^{(l)}$ from Eq. 10) is

$$\tilde{\tau}_{\leq t}^{(l)} = \tilde{\tau}_{\leq t-1}^{(l)} + \tau_t^{(l)}(P_t^{(l)} + B_t^{(l)} A_t^{(l)}). \tag{15}$$

Substituting Eq. 15 into the projection terms yields the compact objective:

$$\mathcal{L}(A_t^{(l)}, B_t^{(l)}) = \left\| \mathcal{T} - (M + \tau_t^{(l)} B_t^{(l)} A_t^{(l)})\hat{V} \right\|_F^2, \tag{16}$$

with

$$\mathcal{T} = \begin{bmatrix} \tilde{\tau}_{\leq t-1}^{(l)} \hat{V}_{\leq t-1}^{(l)} \\ \tau_t^{(l)} \hat{V}_t^{(l)} \end{bmatrix}, \quad \hat{V} = \begin{bmatrix} \hat{V}_{\leq t-1}^{(l)} \\ \hat{\hat{V}}_t^{(l)} \end{bmatrix}, \quad M = \tilde{\tau}_{\leq t-1}^{(l)} + \tau_t^{(l)} P_t^{(l)}. \tag{17}$$

Here the residual $\tau_t^{(l)} B_t^{(l)} A_t^{(l)}$ bridges $M\hat{V}$ and $\mathcal{T}$, ensuring stability to past tasks and plasticity to the new one. Since $\tau_t^{(l)}$ is generally non-square and low-rank, closed-form for $(A_t^{(l)}, B_t^{(l)})$ are unstable; we thus optimize Eq. 16 via gradient descent. The full procedure is summarized in Algo. 1.

## 5 EXPERIMENTS

### 5.1 EXPERIMENTAL SETUPS

**Benchmarks and Protocols.** We evaluate our approach on both vision, NLP tasks, and multimodal tasks to assess scalability and generality. For vision, following (Ilharco et al., 2023; Wang et al.), we adopt CLIP-ViT backbones (Radford et al., 2021) and construct three groups of 8, 14, and 20 image classification tasks. We use publicly available ViT-B/32, ViT-B/16, and ViT-L/14 checkpoints, each fine-tuned on up to 20 datasets (Tang et al., 2024a). To ensure robustness to task order, all experiments are repeated over 10 random permutations (seeds 42–51). For NLP, we evaluate on eight GLUE tasks (Wang et al., 2019) using Flan-T5-base (Chung et al., 2024). For multimodal tasks, we evaluate on LLaVA-1.5-7B (Liu et al., 2023), a widely used vision–language model for instruction following. We adopt four multimodal generative benchmarks: IconQA (Lu et al., 2021), ImageNet captioning (Deng et al., 2009), ScienceQA (Lu et al., 2022), and VizWiz-Caption (Gurari et al., 2018), covering diagram reasoning, natural images, scientific QA, and low-vision real-world scenarios.

**Implementation Details** We insert the null-space filter in a cascaded fashion before selected linear layers of the backbone. All experiments share a single set of *global* hyper-parameters—used across both CLIP and Flan models, as well as all task orders and scales—without task-specific tuning: null-space rank $r_p = 128$, LoRA rank $r_l = 64$, and task projection rank $r_v = 8$. Each task is adapted for 50 iterations with Adam at a learning rate of $1 \times 10^{-3}$.

**Metrics and Baselines.** We evaluate performance with two standard metrics: average accuracy (ACC) and backward transfer (BWT) (Lin et al., 2022). ACC is the mean accuracy of the final merged model across all tasks, $\text{ACC} = \frac{1}{T} \sum_{i=1}^{T} a_i(\theta_T^{\text{merged}})$, where $a_i(\cdot)$ denotes accuracy on task $i$[1]. BWT quantifies the effect of merging on past tasks by comparing their performance before and after the final merge: $\text{BWT} = \frac{1}{T-1} \sum_{i=1}^{T-1} \left[ a_i(\theta_T^{\text{merged}}) - a_i(\theta_i^{\text{merged}}) \right]$. In addition to the methods in Sec. 3.2, we compare against TIES-MERGING (Yadav et al., 2023), MAGMAX-IND (Marczak et al., 2024), WEMOE (Tang et al., 2024b), and WUDI-MERGING (Cheng et al., 2025). Detailed descriptions are provided in the Appendix B.3.

## 5.2 MAIN RESULTS

Table 1: Comparative results of continual merging methods, reporting average accuracy and backward transfer over ten task orders (mean±std). EP and DA denote method assumptions: the need for extra parameters or data access. Best results are in **bold**, and the second best are underlined.

| Method | Requirement EP / DA | ViT-B/32 | | | ViT-B/16 | | | ViT-L/14 | | |
|---|---|---|---|---|---|---|---|---|---|---|
| | | 8 tasks | 14 tasks | 20 tasks | 8 tasks | 14 tasks | 20 tasks | 8 tasks | 14 tasks | 20 tasks |
| PRE-TRAINED | – / – | 48.1 | 56.9 | 55.6 | 55.4 | 62.0 | 59.8 | 64.9 | 69.1 | 65.6 |
| INDIVIDUAL | – / – | 90.4 | 89.3 | 89.8 | 92.4 | 91.3 | 91.6 | 94.3 | 93.4 | 93.5 |
| C. FINE-TUNED | – / – | 79.8 | 67.4 | 62.6 | 82.9 | 72.2 | 68.2 | 90.0 | 70.9 | 77.7 |
| WEIGHT AVERAGING | ✗ / ✗ | 66.3 ±0.0 | 65.4 ±0.0 | 61.1 ±0.0 | 72.3 ±0.0 | 69.7 ±0.0 | 64.8 ±0.0 | 80.0 ±0.0 | 77.5 ±0.0 | 71.1 ±0.0 |
| TASK ARITHMETIC | ✗ / ✗ | 67.5 ±0.0 | 66.5 ±0.0 | 60.0 ±0.0 | 77.1 ±0.0 | 70.9 ±0.6 | 64.2 ±0.0 | 82.1 ±0.0 | 77.9 ±0.0 | 70.3 ±0.0 |
| ← TIES-MERGING | ✗ / ✗ | 49.0 ±10.2 | 66.2 ±0.6 | 59.9 ±0.7 | 66.8 ±3.7 | 70.5 ±0.8 | 63.0 ±1.6 | 64.3 ±7.0 | 78.0 ±0.6 | 68.3 ±0.9 |
| MAGMAX-IND | ✗ / ✗ | 70.7 ±0.0 | 67.0 ±0.0 | 61.2 ±0.0 | 76.7 ±1.8 | 67.0 ±0.0 | 62.5 ±0.0 | 83.4 ±0.0 | 71.2 ±0.0 | 71.2 ±0.0 |
| (%) LW ADAMERGING | ✗ / ✓ | 53.4 ± 3.2 | 59.8 ±1.6 | 59.7 ±7.4 | 59.9 ±2.3 | 64.3 ± 1.2 | 61.5 ±1.1 | 68.8 ±2.9 | 73.1 ±5.7 | 66.9 ± 1.1 |
| CC LoRA-WEMOE | ✓ / ✓ | 68.8 ± 7.8 | 63.8 ±3.4 | 49.6 ±15.4 | 72.6 ±3.7 | 67.9 ±2.9 | 55.0 ±7.0 | 75.6 ± 7.8 | 74.0 ±5.0 | 56.9 ± 19.8 |
| WUDI-MERGING | ✗ / ✗ | 74.7 ±6.6 | 67.0 ±6.9 | 63.7 ±3.8 | 81.0 ±4.7 | 75.0 ±4.1 | 69.6 ±4.7 | 87.5 ±3.3 | 84.2 ±3.7 | 78.1 ±2.8 |
| ISO-C | ✗ / ✗ | 71.7 ±1.2 | 73.2 ±1.8 | 67.6 ±0.8 | 78.5 ±1.2 | 79.7 ±1.3 | 73.0 ±1.1 | 86.9 ±0.5 | 86.9 ±1.8 | 80.9 ±0.8 |
| KNOTS-TIES | ✗ / ✗ | 54.4 ±6.9 | 67.8 ±0.4 | 60.5 ±1.4 | 57.9 ±8.4 | 71.6 ±0.3 | 62.7 ±1.1 | 68.3 ±5.7 | 78.8 ±0.3 | 69.7 ±0.8 |
| TSV-M | ✗ / ✗ | 68.2 ±4.8 | 63.3 ±4.8 | 58.8 ±3.3 | 75.4 ±4.0 | 69.2 ±3.1 | 63.1 ±1.3 | 82.2 ±3.6 | 78.1 ±3.6 | 70.5 ±1.2 |
| OPCM | ✗ / ✗ | 75.5 ±0.5 | 71.9 ±0.3 | 65.7 ±0.2 | 81.8 ±0.3 | 77.1 ±0.5 | 70.3 ±0.2 | 87.0 ±0.4 | 83.5 ±0.2 | 76.0 ±0.2 |
| NUFILT (Ours) | ✗ / ✗ | **83.6 ±0.2** | **78.0 ±0.2** | **71.0 ±0.9** | **87.3 ±0.1** | **83.1 ±0.3** | **78.1 ±0.9** | **91.6 ±0.1** | **89.2 ±0.1** | **84.7 ±0.8** |
| WEIGHT AVERAGING | ✗ / ✗ | -11.5 ±2.2 | -8.0 ±1.3 | -7.1 ±2.1 | -9.7 ±1.5 | -7.1 ±1.4 | -7.3 ±1.7 | -7.3 ±1.4 | -5.8 ±1.0 | -6.4 ±1.5 |
| TASK ARITHMETIC | ✗ / ✗ | -9.6 ±1.5 | -1.3 ±1.6 | -3.4 ±1.0 | -4.2 ±1.0 | -1.3 ±0.4 | -3.6 ±0.4 | -7.1 ±0.8 | -1.8 ±0.3 | -3.3 ±0.3 |
| ← TIES-MERGING | ✗ / ✗ | -15.3 ±8.0 | **1.9 ±0.6** | -1.5 ±0.7 | -5.5 ±0.4 | **1.4 ±0.7** | **-1.5 ±1.2** | -13.0 ±5.7 | -1.1 ±0.4 | -2.9 ±1.0 |
| MAGMAX-IND | ✗ / ✗ | -8.3 ±1.3 | -7.4 ±1.4 | -7.2 ±1.6 | -6.1 ±1.3 | -7.4 ±2.0 | -8.0 ±2.2 | -5.0 ±0.8 | -6.0 ±2.1 | -6.5 ±2.1 |
| (%) LW ADAMERGING | ✗ / ✓ | -32.5 ±3.6 | -24.1 ±1.7 | -22.7 ±4.3 | -27.8 ±2.7 | -22.1 ±1.4 | -21.4 ±1.2 | -24.3 ±3.3 | -19.6 ±1.7 | -21.7 ±1.1 |
| LoRA-WEMOE | ✓ / ✓ | -20.4 ±9.0 | -20.2 ±3.9 | -24.5 ±10.0 | -18.0 ±6.2 | -18.8 ±3.4 | -25.8 ±7.9 | -17.8 ±5.9 | -16.8 ±5.3 | -27.9 ±17.2 |
| WUDI-MERGING | ✗ / ✗ | -17.0 ±7.5 | -22.8 ±7.3 | -26.0 ±4.1 | -12.6 ±5.4 | -16.9 ±4.4 | -18.5 ±14.2 | -7.3 ±3.7 | -9.4 ±4.0 | -15.8 ±2.9 |
| ISO-C | ✗ / ✗ | -10.2 ±1.2 | -10.4 ±1.9 | -10.3 ±1.4 | -6.7 ±0.5 | -7.1 ±1.1 | -9.9 ±1.7 | -3.7 ±0.6 | -3.7 ±1.7 | -5.5 ±1.3 |
| KNOTS-TIES | ✗ / ✗ | -12.6 ±3.9 | **1.9 ±0.5** | **-1.3 ±0.7** | -13.5 ±5.5 | **1.0 ±0.1** | -2.3 ±0.6 | -11.6 ±3.6 | **0.3 ±0.3** | **-2.3 ±0.7** |
| TSV-M | ✗ / ✗ | -24.0 ±5.6 | -26.7 ±4.9 | -31.7 ±3.4 | -18.5 ±4.6 | -22.7 ±3.1 | -28.2 ±1.8 | -13.0 ±4.0 | -15.6 ±3.9 | -23.3 ±1.3 |
| OPCM | ✗ / ✗ | -6.3 ±1.1 | -6.0 ±1.0 | -7.8 ±1.5 | -4.8 ±0.7 | -5.1 ±1.4 | -6.3 ±2.2 | -2.6 ±1.0 | -4.3 ±0.7 | -6.5 ±1.8 |
| NUFILT (Ours) | ✗ / ✗ | **-2.7 ±0.7** | -5.7 ±0.9 | -8.9 ±2.3 | **-1.6 ±0.5** | -3.5 ±0.6 | -7.1 ±1.9 | **-1.1 ±0.3** | -2.0 ±0.3 | -4.6 ±0.7 |

**Results on Vision Tasks.** The comparative results on vision tasks are summarized in Tab. 1, highlighting that our method, NUFILT, consistently outperforms prior merging techniques across various model architectures and task sequences, all without needing extra parameters or data access. Relative to baselines like OPCM and WUDI-Merging, NUFILT achieves marked improvements in average accuracy while reducing backward transfer. For example, on ViT-B/32 with 8 tasks, it reaches 83.6% accuracy, outperforming OPCM by 8.1% and WUDI-Merging by 8.9%; for 20 tasks, the gains are 5.3% over OPCM. On ViT-L/14, it surpasses OPCM by 4.6% on 8 tasks and 8.7% on 20 tasks, with backward transfer limited to -1.1% versus -2.6% for OPCM on 8 tasks. Overall, NUFILT closes the gap to the individual fine-tuning benchmark, lagging by only 2.7% on ViT-L/14 for 8 tasks, underscoring its robustness in continual merging settings.

**Results on NLP Tasks.** The results for NLP tasks using the Flan-T5-base model on 8 tasks are detailed in Tab. 2, showing that NUFILT excels all merging strategies. Against comparable baselines such as OPCM and WUDI-Merging, NUFILT delivers notable gains in overall performance. It achieves an average accuracy of 83.7%, exceeding WUDI-Merging by 1.5% and OPCM by 3.1%, while limiting backward transfer to -1.5% compared to -3.9% for WUDI-Merging and -2.5% for OPCM. NUFILT narrows the divide with individual fine-tuning, trailing by merely 2.7%, affirming its efficacy in continual merging for language models.

---

[1] For STSB (NLP), we report Spearman's $\rho$, denoted by $a_i(\cdot)$ for notational consistency.

**Results on Multimodal tasks.** The results for multimodal tasks using LLaVA-1.5-7B on four benchmarks are summarized in Tab. 3. Compared with strong baselines, NUFILT attains clear improvements in average performance. It reaches an overall score of 70.5%, surpassing OPCM by 2.9%, underscoring its effectiveness in continually merging multimodal models.

Table 2: Results of continual merging Flan-T5-base models on 8 tasks, ordered alphabetically.

| Method | EP / DA | CoLA | MNLI | MRPC | QNLI | QQP | RTE | SST2 | STSB | ACC ↑ | BWT ↑ |
|---|---|---|---|---|---|---|---|---|---|---|---|
| PRE-TRAINED | – / – | 69.1 | 56.5 | 76.2 | 88.4 | 82.1 | 80.1 | 91.2 | 62.2 | 75.7 | - |
| INDIVIDUAL | – / – | 75.0 | 83.4 | 87.5 | 91.5 | 85.4 | 85.9 | 93.6 | 88.7 | 86.4 | - |
| TASK ARITHMETIC | ✗ / ✗ | 69.1 | 58.1 | 77.9 | 88.9 | 83.1 | 79.1 | 90.7 | 74.0 | 77.6 | -4.6 |
| TIES-MERGING | ✗ / ✗ | 39.3 | 70.0 | **82.4** | 88.8 | 81.8 | 75.8 | 89.7 | 76.8 | 75.6 | -6.1 |
| LW ADAMERGING | ✗ / ✓ | 69.1 | 58.1 | 77.9 | 88.9 | 83.1 | 79.1 | 90.7 | 74.2 | 77.6 | -4.7 |
| LORA-WEMOE | ✓ / ✓ | 71.5 | 80.6 | 78.2 | 90.3 | 82.7 | 80.5 | 91.3 | 76.2 | 81.4 | **0.1** |
| WUDI-MERGING | ✗ / ✗ | 71.9 | 73.4 | 79.2 | 89.7 | 82.9 | 79.1 | 93.1 | **88.2** | 82.2 | -3.9 |
| OPCM | ✗ / ✗ | 69.9 | 72.9 | 78.7 | 90.3 | 83.8 | **83.0** | 92.2 | 73.7 | 80.6 | -2.5 |
| NUFILT (Ours) | ✗ / ✗ | **72.5** | **83.3** | 78.7 | **91.1** | **84.2** | 79.4 | **93.4** | 87.1 | **83.7** | -1.5 |

Table 3: Results of continual merging LLaVA models on 4 tasks, ordered alphabetically.

| Method | IconQA | Image | SciQA | VizWiz | Average |
|---|---|---|---|---|---|
| INDIVIDUAL | 75.5 | 96.0 | 83.7 | 64.8 | 80.0 |
| PRE-TRAINED | 14.1 | 40.9 | 61.7 | 41.0 | 39.4 |
| TASK ARITHMETIC | 47.6 | 67.9 | 74.2 | 46.6 | 59.1 |
| TIES-MERGING | **69.6** | 74.4 | 75.5 | 44.6 | 66.0 |
| OPCM | 58.0 | 86.4 | **81.1** | 44.9 | 67.6 |
| NUFILT (Ours) | 59.6 | **93.6** | 80.6 | **48.3** | **70.5** |

## 5.3 ABLATION AND ANALYSIS RESULTS

**Ablation on Each Component.** We conduct an ablation study to disentangle the contributions of the null-space filter and the data-free objective. Results are summarized in Tab. 4. (1) Simply accumulating task vectors without any constraint leads to severe performance degradation. Both accuracy and backward transfer drop sharply as the number of tasks increases, confirming that direct parameter addition suffers from catastrophic interference. (2) Projecting task vectors onto the null-space of previous tasks significantly stabilizes merging. Compared to naive merging, this strategy improves accuracy by more than 20 points across all task counts and reduces forgetting to near zero. (3) Using the data-free objective alone (without null-space filtering) recovers part of the performance but still suffers from negative backward transfer, especially when scaling to more tasks. Combining both components yields the best balance between stability and plasticity.

Table 4: Ablation study of NUFILT with CLIP ViT-B/32 over 8, 14, and 20 tasks.

| Method | Component | | ACC(%) ↑ | | | BWT(%) ↑ | | |
|---|---|---|---|---|---|---|---|---|
| | Null-space / LoRA | | 8 tasks | 14 tasks | 20 tasks | 8 tasks | 14 tasks | 20 tasks |
| **(1) Naive Merging** ($\theta_t^{\mathrm{merged}} = \theta_{t-1}^{\mathrm{merged}} + \tau_t$) | ✗ / ✗ | | $62.1_{\pm 0.0}$ | $46.5_{\pm 0.0}$ | $34.3_{\pm 0.0}$ | $-18.5_{\pm 6.2}$ | $-25.8_{\pm 2.2}$ | $-24.7_{\pm 5.1}$ |
| **(2) Only Null-space** ($\theta_t^{\mathrm{merged}} = \theta_{t-1}^{\mathrm{merged}} + \tau_t P_t$) | ✓ / ✗ | | $80.0_{\pm 1.1}$ | $76.7_{\pm 1.0}$ | $67.0_{\pm 0.7}$ | $-1.7_{\pm 0.9}$ | $-4.2_{\pm 0.9}$ | $-6.2_{\pm 1.8}$ |
| **(3) Data-free Objective** | | | | | | | | |
| w/o null-space filter ($\theta_t^{\mathrm{merged}} = \theta_{t-1}^{\mathrm{merged}} + \tau_t B_t A_t$) | ✗ / ✓ | | $75.8_{\pm 1.9}$ | $63.7_{\pm 1.6}$ | $51.7_{\pm 1.0}$ | $-10.2_{\pm 4.3}$ | $-17.1_{\pm 2.7}$ | $-20.6_{\pm 4.9}$ |
| full method ($\theta_t^{\mathrm{merged}} = \theta_{t-1}^{\mathrm{merged}} + \tau_t(P_t + B_t A_t)$) | ✓ / ✓ | | $83.6_{\pm 0.2}$ | $78.0_{\pm 0.2}$ | $71.0_{\pm 0.9}$ | $-2.7_{\pm 0.7}$ | $-5.7_{\pm 0.9}$ | $-8.9_{\pm 2.3}$ |

**Hyper-parameter Sensitivity.** We analyze the impact of rank hyper-parameters in Fig. 4. (a) Higher null-space rank $r_p$ initially reduces forgetting and boosts performance, but excessively large ranks reduce projection selectivity, impairing new task adaptation and lowering accuracy. (b) Low-rank adaptation enhances adaptability while controlling forgetting. Increasing rank $r_l$ improves accuracy steadily, with backward transfer remaining stable. (c) Projecting updates onto $\hat{V}_t^{(l)}$ improves new task adaptability. Moderate ranks yield consistent gains, while overly large $r_v$ reduces projection specificity, approaching full-space projection and diminishing benefits.

**Evaluation of stability and plasticity.** Following Eqs. 3 and 4, we compute $\mathcal{L}_{\mathrm{stab}}^{(l)}$ and $\mathcal{L}_{\mathrm{plas}}^{(l)}$ layer-wise and report their averages over all layers. $\mathcal{L}_{\mathrm{stab}}^{(l)}$ measures activation changes on earlier tasks, and $\mathcal{L}_{\mathrm{plas}}^{(l)}$

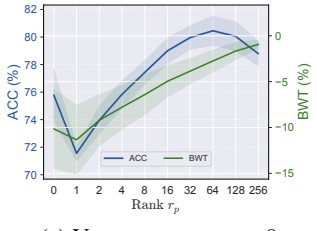 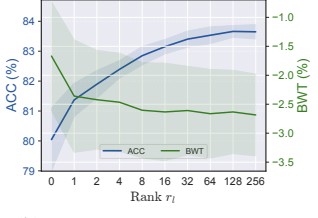 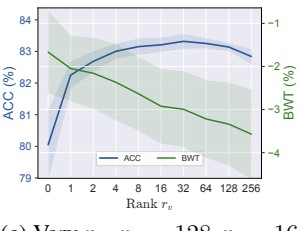

(a) Vary $r_p$, $r_l = r_v = 0$  (b) Vary $r_l$, $r_p = 128$, $r_v = 8$  (c) Vary $r_v$, $r_p = 128$, $r_l = 16$

Figure 4: Hyper-parameter sensitivity analysis on the 8-task continual merging protocol. Setting $r = 0$ corresponds to removing the associated component.

measures deviation from the individual model of the current task $t$. Since both are losses, lower values indicate better stability and plasticity. As shown in Fig. 5, NUFILT achieves much lower stability loss, indicating minimal disturbance to earlier tasks, and consistently lower plasticity loss, showing closer alignment with the individual solution. The harmonic mean further confirms that NUFILT achieves the best balance between the two criteria.

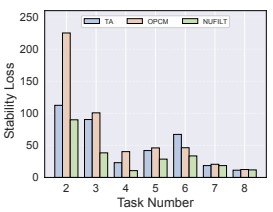 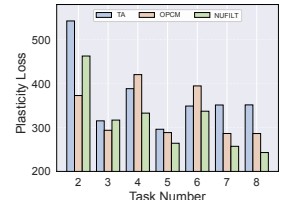 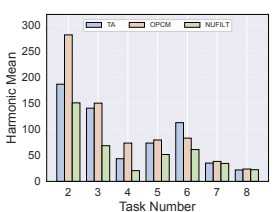

Figure 5: Evaluation of stability and plasticity surrogates, and their harmonic mean on the CLIP ViT-B/32 8-task continual merging protocol. Lower values in all metrics indicate better performance.

**Computation Overhead.** Tab. 5 compares the computation overhead of three categories of methods. Training-free approaches (TA, OPCM) are the most efficient since they require no optimization or data loading, but their accuracy is limited. Test-time methods (LW AdaMerging, LoRA-WEMOE) exhibit the largest overall runtime because both data loading and multi-step training add substantial latency. Compared with WUDI-Merging, NUFILT achieves lower solving time because its LoRA-based updates operate on fewer parameters, though its total time is slightly higher owing to the additional SVD computation. Overall, NUFILT provides the best accuracy while maintaining competitive efficiency.

Table 5: Overhead of ViT-B/32 under 8-task continual merging: per-task iterations, solving time, total time, peak GPU memory, and final parameter count.

| Model | Iters. | Solv. | Tot. | GPU Mem. | Param. | ACC(%) |
|---|---|---|---|---|---|---|
| TA | - | - | 8.3s | - | 87.5M | $67.5_{\pm 0.0}$ |
| OPCM | - | - | 76.6s | 1.3GB | 87.5M | $75.5_{\pm 0.5}$ |
| LW ADAMERGING | 50 | 47.1s | 724.2s | 1.6GB | 87.5M | $53.4_{\pm 3.2}$ |
| LoRA-WEMOE | 50 | 72.9s | 754.6s | 1.8GB | 103.7M | $68.8_{\pm 7.8}$ |
| WUDI-MERGING | 50 | 28.9s | 37.7s | 2.0GB | 87.5M | $74.7_{\pm 6.6}$ |
| | 25 | 9.3s | 129.8s | 1.8GB | 87.5M | $83.3_{\pm 0.3}$ |
| NUFILT | 100 | 35.9s | 156.4s | 1.8GB | 87.5M | $83.6_{\pm 0.2}$ |
| | 50 | 18.4s | 138.9s | 1.8GB | 87.5M | $83.6_{\pm 0.2}$ |

## 6  CONCLUSION

In this work, we addressed the problem of data-free continual model merging, where independently fine-tuned models must be consolidated sequentially into a single backbone without access to task data or earlier checkpoints. The central challenge lies in enforcing stability and plasticity. To this end, we established that task vectors approximately align with representation subspaces, providing a geometric foundation for continual merging. Building on this insight, we proposed NUFILT, a novel framework that integrates null-space filtering to suppress interference with prior knowledge and projection-aware adaptation to recover task-specific plasticity via data-free surrogate objectives. Experiments across vision and NLP benchmarks demonstrated that NUFILT achieves state-of-the-art results, improving over recent methods by 4–7% on average while substantially reducing forgetting and narrowing the gap to individual fine-tuning. We believe this work provides a principled and practical step toward scalable, theoretically grounded solutions for continual model merging.

**Acknowledgments** This work was supported in part by National Science and Technology Major Project (2021ZD0112001), National Natural Science Foundation of China (No.62271119, 08120002, U23A20286, 62071086), the Key Research and Development Project of Hainan Province (Grant No. ZDYF2024(LALH)003), Liaoning Province Grant XLYC2403174, the Natural Science Foundation of Sichuan Province under Grant 2025ZNSFSC0475.

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

APPENDIX

The appendix is organized into sections, each providing supplementary explanations and supporting details.

# A   PROOF OF THEORETICAL RESULTS

## A.1   PROOF OF THEOREM 1

Theorem 1 extends Proposition A.1 (Cheng et al., 2025) from individual task vectors to subspaces. While Proposition A.1 demonstrates that each task vector can be approximated as a linear combination of representation features, Theorem 1 further establishes approximate alignment between the singular subspaces of task vectors and task representations.

**Theorem A.1** (Weyl inequality for singular values (Weyl, 1912)). *For matrices $A \in \mathbb{R}^{m \times n}$ and $E \in \mathbb{R}^{m \times n}$, with singular values $\sigma_j(A)$ and $\sigma_j(A+E)$ ordered decreasingly ($\sigma_1 \geq \sigma_2 \geq \cdots \geq 0$), and $\|E\|_2$ the operator norm of $E$, the following holds for $j = 1, 2, \ldots, \min(m, n)$:*

$$|\sigma_j(A + E) - \sigma_j(A)| \leq \|E\|_2. \tag{18}$$

**Theorem A.2** (Wedin's Sin-Theta Theorem (Wedin, 1972)). *Let $M \in \mathbb{R}^{m \times n}$ with SVD $M = U\Sigma V^\top, U = [U_0 \quad U_\perp], V = [V_0 \quad V_\perp]$, where $U_0 \in \mathbb{R}^{m \times r}$ and $V_0 \in \mathbb{R}^{n \times r}$ correspond to the top-$r$ singular values. Let $\hat{M} = M + H$ with SVD $\hat{M} = \hat{U}\hat{\Sigma}\hat{V}^\top, \hat{U} = \begin{bmatrix} \hat{U}_0 & \hat{U}_\perp \end{bmatrix}, \hat{V} = \begin{bmatrix} \hat{V}_0 & \hat{V}_\perp \end{bmatrix}$. Suppose there exist $a \in \mathbb{R}$ and $\Delta > 0$ such that $\sigma_r(M) \geq a, \sigma_{r+1}(\hat{M}) \leq a - \Delta$. Then the canonical angles between the singular subspaces satisfy*

$$\max\left\{ \mathrm{dist}(\hat{U}_0, U_0), \ \mathrm{dist}(\hat{V}_0, V_0) \right\} \leq \frac{\max\{\|HV_0\|_2, \|H^\top U_0\|_2\}}{\Delta} \leq \frac{\|H\|_2}{\Delta}, \tag{19}$$

*where $\mathrm{dist}(\hat{U}_0, U_0) = \|\sin\Theta(\hat{U}_0, U_0)\|_2$ denotes the largest principal angle between subspaces.*

**Proposition A.1** (Approximate Linear Combination (Cheng et al., 2025)). *Let $\tau_k^{(l)}$ denote the task vector of neuron $k$ in linear layer $l$. Consider $N$ input samples $\{x_n\}_{n=1}^N$, and let $x_{n,t}^{(l)}$ denote the corresponding input to layer $l$ after $t$ iteration for sample $x_n$ during fine-tuning. Assume that the model before layer $l$ is $C_l$-Lipschitz continuous, and the gradient of loss is bounded in $\ell_2$-norm by $\mathcal{G}_l$, and the gradient of the loss with respect to the product $\theta_{k,t-1}^{(l)} x_{n,t}^{(l)}$ is bounded by $\Gamma_k^{(l)}$. Then, the following inequality holds:*

$$\left\| \tau_k^{(l)} - \sum_{n=1}^N \beta_{k,n}^{(l)} (x_{n,T}^{(l)})^\top \right\|_2 \le \Phi_k \cdot \left( \sum_{t=1}^T \sum_{i=t}^T \eta_t \eta_i \right). \tag{20}$$

*where $\Phi_k = N \cdot C_l \cdot \mathcal{G}_l \cdot \Gamma_k^{(l)}$ and $\beta_{k,n}^{(l)} = \sum_{t=1}^T -\eta_t \frac{\partial L(\theta_t)}{\partial(\theta_{k,t-1}^{(l)} x_{n,t}^{(l)})}$.*

**Theorem A.3** (Approximate Subspace Alignment). *Let $\tau^{(l)} \in \mathbb{R}^{d_o \times d_i}$ be the task vector at layer $l$, and $H \in \mathbb{R}^{N \times d_i}$ a representation matrix of rank $r_d$ with right singular vectors $V_d \in \mathbb{R}^{d_i \times r_d}$. Suppose $\tau^{(l)}$ admits the decomposition $\tau^{(l)} = T_0 + E$ with $T_0 = BH$, $\mathrm{rank}(T_0) = r_d$, where each row $E_k$ of $E$ satisfies $\|E_k\|_2 \le \Psi_k$. Let $\sigma_{r_d}(T_0)$ denote the $r_d$-th largest singular value of the matrix $T_0$. If $\sigma_{r_d}(T_0) > \sqrt{d_o} \max_k \Psi_k$, then for any $r_v \ge r_d$, the span of the top $r_v$ right singular vectors $\hat{V}$ of $\tau^{(l)}$ is approximately aligned with $\mathrm{span}(V_d)$ in the sense that*

$$1 - \mathcal{A}(V_d^{(l)}, \hat{V}^{(l)}) \le \zeta^2, \tag{21}$$

*where $\mathcal{A}(V_d^{(l)}, \hat{V}^{(l)}) = \frac{1}{r_d} \|\hat{V}^\top V_d\|_F^2$ and $\zeta = \frac{\sqrt{d_o} \max_k \Psi_k}{\sigma_{r_d}(T_0) - \sqrt{d_o} \max_k \Psi_k}$.*

*Proof.* The proof of Theorem 1 requires three ingredients. First, Proposition A.1 shows that task vectors can be approximated by linear combinations of representation features, yielding a decomposition into a structured term plus bounded error. Second, Weyl's inequality (Theorem A.1) provides a perturbation bound for singular values, which ensures a nontrivial spectral gap under bounded error. Finally, Wedin's sin–Theta theorem (Theorem A.2) translates this spectral gap into a guarantee of approximate alignment between the true and perturbed subspaces.

From Proposition A.1 (Approximate Linear Combination), each row $\tau_k^{(l)}$ of the task vector matrix satisfies

$$\|\tau_k^{(l)} - \sum_{n=1}^N \beta_{k,n}^{(l)} (x_{n,T}^{(l)})^\top\|_2 \le \Psi_k, \tag{22}$$

where $\Psi_k = \Phi_k \cdot \left( \sum_{t=1}^T \sum_{i=t}^T \eta_t \eta_i \right)$ is a per-neuron error bound, and $\Phi_k$ depends on the number of samples, Lipschitz constant, gradient bounds, and learning rates. This yields the decomposition $\tau^{(l)} = T_0 + E$, with $T_0 = BH$, $B_{k,n} = \beta_{k,n}^{(l)}$, and $\|E_k\|_2 \le \Psi_k$ for each row $E_k$, so $\|E\|_2 \le \sqrt{d_o} \max_k \Psi_k$.

**(1) Exact case ($E = 0$).** In this case, $\tau^{(l)} = T_0 = BH$. Since $H$ has rank $r_d$, the right singular space of $H$ is exactly $\mathrm{span}(V_d)$. Under the rank assumption $\mathrm{rank}(T_0) = r_d$ (which holds when $d_o \ge r_d$ and $B$ has sufficient rank, as motivated by the NTK regime), $T_0$ and $H$ share the same $r_d$-dimensional right singular subspace. Hence $\mathrm{span}(\hat{V}_{r_d}) = \mathrm{span}(V_d)$, so for any $r_v \ge r_d$, $\mathcal{A}(V_d, \hat{V}) = 1$.

**(2) Perturbed case ($E \ne 0$).** Apply Wedin's $\sin \Theta$ theorem A.2. Let $V_0$ be the top $r_d$ right singular vectors of $T_0$, and set $a = \sigma_{r_d}(T_0)$. Define $\Delta = \sigma_{r_d}(T_0) - \|E\|_2$. The stability assumption $\sigma_{r_d}(T_0) > \sqrt{d_o} \max_k \Psi_k \ge \|E\|_2$ ensures $\Delta > 0$. With $\mathrm{rank}(T_0) = r_d$ and $\tau^{(l)} = T_0 + E$, Weyl's inequality A.1 gives $|\sigma_{r_d+1}(\tau^{(l)}) - \sigma_{r_d+1}(T_0)| \le \|E\|_2$. Since $\sigma_{r_d+1}(T_0) = 0$, this yields $\sigma_{r_d+1}(\tau^{(l)}) \le \|E\|_2 = a - \Delta$.

By Wedin's theorem, the canonical angles $\Theta$ between $\text{span}(V_0)$ and $\text{span}(\hat{V}_{r_d})$ satisfy

$$\| \sin \Theta(V_0, \hat{V}_{r_d}) \|_2 \leq \frac{\|E\|_2}{\Delta}. \tag{23}$$

We have $\text{span}(V_0) = \text{span}(V_d)$ and $\|E\|_2 \leq \sqrt{d_o} \max_k \Psi_k$, hence

$$\| \sin \Theta(V_d, \hat{V}_{r_d}) \|_2 \leq \frac{\sqrt{d_o} \max_k \Psi_k}{\sigma_{r_d}(T_0) - \sqrt{d_o} \max_k \Psi_k}. \tag{24}$$

The subspace affinity then obeys

$$\mathcal{A}(V_d, \hat{V}_{r_d}) = \frac{1}{r_d} \sum_{j=1}^{r_d} \cos^2 \theta_j \ \geq \ 1 - \sin^2 \theta_{\max}. \tag{25}$$

Thus, we obtain

$$1 - \mathcal{A}(V_d, \hat{V}_{r_d}) \ \leq \ \sin^2 \theta_{\max} \ \leq \ \left( \frac{\sqrt{d_o} \max_k \Psi_k}{\sigma_{r_d}(T_0) - \sqrt{d_o} \max_k \Psi_k} \right)^2 = \zeta^2. \tag{26}$$

Finally, since enlarging the subspace only increases the overlap,

$$\mathcal{A}(V_d, \hat{V}) \geq \mathcal{A}(V_d, \hat{V}_{r_d}). \tag{27}$$

Hence, for any $r_v \geq r_d$,

$$1 - \mathcal{A}(V_d, \hat{V}) \ \leq \ 1 - \mathcal{A}(V_d, \hat{V}_{r_d}) \ \leq \ \zeta^2. \tag{28}$$

$\square$

## A.2 Proof of Corollary 1

> **Corollary A.1** (Data-free upper bound). *Let $X \in \mathbb{R}^{N \times d_i}$ with largest singular value $\sigma_1(X)$, rank $r_d$, and right singular vectors $V_d$. Let $\tau \in \mathbb{R}^{d_o \times d_i}$ with top-$r_v$ right singular vectors $\hat{V}$ ($r_v \geq r_d$). If Theorem 1 ensures $1 - \frac{1}{r_d} \|\hat{V}^\top V_d\|_F^2 \ \leq \ \zeta^2$, then for any $\rho \in \mathbb{R}^{d_o \times d_i}$,*
>
> $$\left\| (\rho - \tau) X^\top \right\|_F^2 \ \leq \ 2 \sigma_1(X)^2 \Big( \left\| (\rho - \tau) \hat{V} \right\|_F^2 + r_d \zeta^2 \|\rho - \tau\|_2^2 \Big). \tag{29}$$

*Proof.* Let $P = \hat{V} \hat{V}^\top$ be the projection onto $\text{span}(\hat{V})$. Decompose

$$X^\top = P X^\top + (I - P) X^\top. \tag{30}$$

By $\|a + b\|_F^2 \leq 2\|a\|_F^2 + 2\|b\|_F^2$,

$$\|(\rho - \tau) X^\top\|_F^2 \ \leq \ 2\|(\rho - \tau) P X^\top\|_F^2 + 2\|(\rho - \tau)(I - P) X^\top\|_F^2. \tag{31}$$

**(1) Projection term.**

$$\|(\rho - \tau) P X^\top\|_F^2 = \text{trace}\big((\rho - \tau) P X^\top X P (\rho - \tau)^\top\big)$$
$$= \text{trace}\big(P(\rho - \tau)^\top (\rho - \tau) P X^\top X\big).$$

Let $B = P(\rho - \tau)^\top (\rho - \tau) P \succeq 0$ and $C = X^\top X \succeq 0$. Since $C \preceq \|C\|_2 I$, it follows that $\text{trace}(BC) \leq \|C\|_2 \text{trace}(B)$. Since $\|C\|_2 = \|X^\top X\|_2 = \sigma_1^2(X)$ and

$$\text{trace}(B) = \text{trace}\big(P(\rho - \tau)^\top (\rho - \tau) P\big) = \|(\rho - \tau) P\|_F^2 = \|(\rho - \tau)\hat{V}\|_F^2, \tag{32}$$

we obtain

$$\|(\rho - \tau) P X^\top\|_F^2 \ \leq \ \sigma_1^2(X) \, \|(\rho - \tau)\hat{V}\|_F^2. \tag{33}$$

**(2) Residual term.** By $\|AB\|_F \leq \|A\|_2 \|B\|_F$,

$$\|(\rho - \tau)(I - P)X^\top\|_F^2 \leq \|\rho - \tau\|_2^2 \|X(I - P)\|_F^2. \tag{34}$$

Using the SVD $X = U_d \Sigma_d V_d^\top$,

$$\|X(I - P)\|_F^2 = \text{trace}\big(\Sigma_d^2 V_d^\top (I - P) V_d\big). \tag{35}$$

Since $\Sigma_d^2 \preceq \sigma_1^2(X)I$,

$$\|X(I - P)\|_F^2 \leq \sigma_1^2(X) \text{trace}\big(V_d^\top (I - P) V_d\big). \tag{36}$$

Noting that

$$\text{trace}(V_d^\top (I - P) V_d) = r_d - \|V_d^\top \hat{V}\|_F^2 \leq r_d \zeta^2, \tag{37}$$

we conclude

$$\|X(I - P)\|_F^2 \leq \sigma_1^2(X) r_d \zeta^2. \tag{38}$$

Hence

$$\|(\rho - \tau)(I - P)X^\top\|_F^2 \leq \sigma_1^2(X) r_d \zeta^2 \|\rho - \tau\|_2^2. \tag{39}$$

**(3) Combine.** Summing the two parts yields

$$\|(\rho - \tau)X^\top\|_F^2 \leq 2\sigma_1^2(X) \|(\rho - \tau)\hat{V}\|_F^2 + 2\sigma_1^2(X) r_d \zeta^2 \|\rho - \tau\|_2^2, \tag{40}$$

as claimed. $\qquad\qquad\qquad\qquad\qquad\qquad\qquad\qquad\qquad\qquad\qquad\qquad\qquad\qquad\square$

# B ADDITIONAL DESCRIPTIONS

## B.1 DETAILS OF DATASET AND TASK SETTINGS

**Overview of Vision Tasks** To comprehensively examine continual model merging, we curated a collection of 20 diverse image classification benchmarks, spanning natural objects, remote sensing, medical imagery, and text-rendered datasets. This selection largely follows prior practice in multi-domain evaluation (Tang et al., 2025), while ensuring balanced inclusion of datasets with varying granularity (from binary recognition to hundreds of categories). Specifically, the benchmarks include: SUN397 (Xiao et al., 2010), Stanford Cars (Krause et al., 2013), RESISC45 (Cheng et al., 2017), EuroSAT (Helber et al., 2019), SVHN (Netzer et al., 2011), GTSRB (Stallkamp et al., 2012), MNIST (LeCun et al., 1998), DTD (Cimpoi et al., 2014), Flowers102 (Nilsback & Zisserman, 2008), PCAM (Veeling et al., 2018), FER2013 (Goodfellow et al., 2013), Oxford-IIIT Pet (Parkhi et al., 2012), STL-10 (Coates et al., 2011), CIFAR-100 and CIFAR-10 (Krizhevsky & Hinton, 2009), Food-101 (Bossard et al., 2014), Fashion-MNIST (Xiao et al., 2017), EMNIST (Cohen et al., 2017), KMNIST (Clanuwat et al., 2018), and Rendered SST-2 (Socher et al., 2013).

**Overview of NLP Tasks** In addition to vision benchmarks, we also consider widely-used natural language understanding datasets. Specifically, we evaluate on the GLUE benchmark (Wang et al., 2018), which covers tasks ranging from single-sentence acceptability judgments to pairwise entailment and semantic similarity. For classification-oriented datasets, we report *exact match accuracy*, including CoLA (linguistic acceptability), MNLI (natural language inference), MRPC (paraphrase detection), QNLI (question–answer entailment), QQP (duplicate question detection), RTE (recognizing textual entailment), and SST-2 (sentiment classification). For STS-B, which measures semantic textual similarity, performance is reported using *Spearman's $\rho$ correlation coefficient*.

**Task Grouping** We evaluate continual merging under three progressively larger task sets. For each setting, models are assessed using *average accuracy (ACC)* and *backward transfer (BWT)*. To ensure robustness, we generate 10 random task orders for every group (Tab. 6), reporting the mean and standard deviation across runs.

- **8-Task Group**: (1) SUN397, (2) Stanford Cars, (3) RESISC45, (4) EuroSAT, (5) SVHN, (6) GTSRB, (7) MNIST, (8) DTD.
- **14-Task Group**: Extends the 8-task set with (9) Flowers102, (10) PCAM, (11) FER2013, (12) Oxford-IIIT Pet, (13) STL-10, (14) CIFAR-100.

- **20-Task Group**: Builds upon the 14-task set by adding (15) CIFAR-10, (16) Food-101, (17) Fashion-MNIST, (18) EMNIST, (19) KMNIST, (20) Rendered SST-2.

For the NLP benchmarks, tasks are organized in **alphabetical order**: CoLA, MNLI, MRPC, QNLI, QQP, RTE, SST-2, and STS-B.

Table 6: Dataset orderings used for vision experiments in each task group.

| | Order | Dataset Order (by ID) |
|---|---|---|
| **8 tasks** | 1 | $(04 \to 05 \to 07 \to 08 \to 03 \to 06 \to 01 \to 02)$ |
| | 2 | $(07 \to 08 \to 05 \to 04 \to 02 \to 06 \to 03 \to 01)$ |
| | 3 | $(03 \to 06 \to 04 \to 02 \to 01 \to 08 \to 05 \to 07)$ |
| | 4 | $(06 \to 08 \to 02 \to 01 \to 03 \to 07 \to 04 \to 05)$ |
| | 5 | $(07 \to 06 \to 03 \to 08 \to 05 \to 01 \to 04 \to 02)$ |
| | 6 | $(07 \to 02 \to 03 \to 08 \to 05 \to 04 \to 01 \to 06)$ |
| | 7 | $(07 \to 01 \to 04 \to 03 \to 08 \to 05 \to 02 \to 06)$ |
| | 8 | $(08 \to 05 \to 06 \to 07 \to 01 \to 04 \to 03 \to 02)$ |
| | 9 | $(01 \to 04 \to 05 \to 02 \to 06 \to 03 \to 07 \to 08)$ |
| | 10 | $(08 \to 03 \to 01 \to 02 \to 06 \to 05 \to 07 \to 04)$ |
| **14 tasks** | 1 | $(09 \to 13 \to 08 \to 07 \to 14 \to 12 \to 06 \to 03 \to 10 \to 04 \to 05 \to 01 \to 02 \to 11)$ |
| | 2 | $(09 \to 10 \to 11 \to 14 \to 07 \to 13 \to 04 \to 02 \to 06 \to 08 \to 03 \to 12 \to 05 \to 01)$ |
| | 3 | $(05 \to 08 \to 12 \to 06 \to 11 \to 01 \to 10 \to 04 \to 14 \to 03 \to 02 \to 13 \to 09 \to 07)$ |
| | 4 | $(03 \to 10 \to 09 \to 12 \to 04 \to 13 \to 01 \to 06 \to 11 \to 02 \to 14 \to 08 \to 07 \to 05)$ |
| | 5 | $(08 \to 14 \to 09 \to 06 \to 12 \to 13 \to 05 \to 03 \to 04 \to 11 \to 10 \to 01 \to 07 \to 02)$ |
| | 6 | $(03 \to 12 \to 13 \to 01 \to 11 \to 04 \to 10 \to 05 \to 14 \to 08 \to 09 \to 07 \to 02 \to 06)$ |
| | 7 | $(07 \to 01 \to 12 \to 10 \to 02 \to 08 \to 13 \to 04 \to 05 \to 11 \to 14 \to 03 \to 06 \to 09)$ |
| | 8 | $(05 \to 12 \to 04 \to 11 \to 03 \to 08 \to 10 \to 01 \to 09 \to 13 \to 14 \to 07 \to 06 \to 02)$ |
| | 9 | $(10 \to 07 \to 09 \to 02 \to 03 \to 13 \to 01 \to 12 \to 14 \to 04 \to 11 \to 06 \to 05 \to 08)$ |
| | 10 | $(01 \to 02 \to 11 \to 06 \to 08 \to 12 \to 07 \to 05 \to 10 \to 14 \to 03 \to 13 \to 09 \to 04)$ |
| **20 tasks** | 1 | $(20 \to 06 \to 15 \to 05 \to 10 \to 14 \to 16 \to 19 \to 07 \to 13 \to 18 \to 11 \to 02 \to 12 \to 03 \to 17 \to 08 \to 09 \to 01 \to 04)$ |
| | 2 | $(09 \to 14 \to 06 \to 03 \to 07 \to 04 \to 18 \to 01 \to 17 \to 19 \to 08 \to 20 \to 13 \to 16 \to 11 \to 12 \to 15 \to 05 \to 10 \to 02)$ |
| | 3 | $(09 \to 15 \to 16 \to 11 \to 03 \to 13 \to 08 \to 10 \to 12 \to 02 \to 20 \to 01 \to 05 \to 19 \to 07 \to 06 \to 04 \to 18 \to 17 \to 14)$ |
| | 4 | $(17 \to 04 \to 11 \to 19 \to 18 \to 10 \to 07 \to 15 \to 12 \to 13 \to 08 \to 02 \to 01 \to 06 \to 05 \to 03 \to 20 \to 16 \to 14 \to 09)$ |
| | 5 | $(14 \to 16 \to 04 \to 20 \to 15 \to 17 \to 07 \to 11 \to 06 \to 18 \to 12 \to 01 \to 19 \to 09 \to 10 \to 05 \to 08 \to 02 \to 13 \to 03)$ |
| | 6 | $(02 \to 06 \to 17 \to 04 \to 19 \to 18 \to 08 \to 16 \to 20 \to 01 \to 10 \to 13 \to 07 \to 09 \to 05 \to 11 \to 15 \to 14 \to 03 \to 12)$ |
| | 7 | $(19 \to 01 \to 09 \to 14 \to 06 \to 20 \to 17 \to 04 \to 08 \to 02 \to 15 \to 03 \to 16 \to 13 \to 12 \to 07 \to 10 \to 05 \to 11 \to 18)$ |
| | 8 | $(15 \to 07 \to 08 \to 02 \to 10 \to 06 \to 17 \to 20 \to 05 \to 19 \to 16 \to 01 \to 18 \to 09 \to 13 \to 11 \to 04 \to 14 \to 12 \to 03)$ |
| | 9 | $(10 \to 05 \to 07 \to 11 \to 01 \to 03 \to 17 \to 15 \to 18 \to 04 \to 14 \to 19 \to 02 \to 06 \to 13 \to 20 \to 08 \to 12 \to 09 \to 16)$ |
| | 10 | $(01 \to 11 \to 02 \to 15 \to 03 \to 10 \to 12 \to 19 \to 16 \to 13 \to 07 \to 05 \to 09 \to 04 \to 14 \to 20 \to 06 \to 18 \to 17 \to 08)$ |

## B.2 DETAILS OF DOWNSTREAM MODELS

In this section, we describe the evaluation protocol for both pre-trained and fine-tuned models across vision and natural language processing (NLP) downstream tasks.

For vision tasks (Tab. 7), we report the zero-shot performance of pre-trained CLIP-ViT models as well as the test accuracy of task-specific fine-tuned models. Fine-tuned checkpoints are obtained from Hugging Face (`https://huggingface.co/tanganke`), where each model is trained on its respective dataset using a standard recipe. During fine-tuning, the visual encoder is updated while the text encoder remains fixed. Training follows a consistent setup: cross-entropy loss, Adam optimizer, cosine annealing schedule, learning rate of $1 \times 10^{-5}$, batch size of 128, and 4000 training steps.

For NLP tasks, we adopt the 8-task GLUE benchmark using Flan-T5-base. As summarized in Tab. 8, we compare pre-trained and fine-tuned Flan-T5-base models across all GLUE tasks. Fine-tuning is conducted with learning rate $4 \times 10^{-5}$, batch size 16, and 2000 training steps for each task.

Table 7: Performance of the CLIP pre-trained model and individually fine-tuned models on different vision downstream tasks.

| | Model | SUN397 | Cars | RESISC45 | EuroSAT | SVHN | GTSRB | MNIST | DTD | Flowers102 | PCAM |
|---|---|---|---|---|---|---|---|---|---|---|---|
| Pre-trained | CLIP-ViT-B/32 | 63.2 | 59.6 | 60.3 | 45.0 | 31.6 | 32.5 | 48.3 | 44.2 | 66.4 | 60.6 |
| | CLIP-ViT-B/16 | 65.5 | 64.7 | 66.4 | 54.1 | 52.0 | 43.5 | 51.7 | 45.0 | 71.3 | 54.0 |
| | CLIP-ViT-L/14 | 68.2 | 77.9 | 71.3 | 61.2 | 58.4 | 50.5 | 76.3 | 55.5 | 79.2 | 51.2 |
| Fine-tuned | CLIP-ViT-B/32 | 74.9 | 78.5 | 95.1 | 99.1 | 97.3 | 98.9 | 99.6 | 79.7 | 88.6 | 88.0 |
| | CLIP-ViT-B/16 | 78.9 | 85.9 | 96.6 | 99.0 | 97.6 | 99.0 | 99.7 | 82.3 | 94.9 | 90.6 |
| | CLIP-ViT-L/14 | 82.8 | 92.8 | 97.4 | 99.1 | 97.9 | 99.2 | 99.8 | 85.5 | 97.7 | 91.1 |

| | Model | FER2013 | OxfordIIITPet | STL10 | CIFAR100 | CIFAR10 | Food101 | FashionMNIST | EMNIST | KMNIST | RenderedSST2 |
|---|---|---|---|---|---|---|---|---|---|---|---|
| Pre-trained | CLIP-ViT-B/32 | 41.3 | 83.3 | 97.1 | 63.7 | 89.8 | 82.4 | 63.0 | 12.0 | 10.0 | 58.6 |
| | CLIP-ViT-B/16 | 46.4 | 88.4 | 98.3 | 66.3 | 90.8 | 87.0 | 67.3 | 12.4 | 11.2 | 60.6 |
| | CLIP-ViT-L/14 | 50.0 | 93.2 | 99.4 | 75.1 | 95.6 | 91.2 | 67.0 | 12.3 | 9.7 | 68.9 |
| Fine-tuned | CLIP-ViT-B/32 | 71.6 | 92.5 | 97.5 | 88.4 | 97.6 | 88.4 | 94.7 | 95.6 | 98.2 | 71.3 |
| | CLIP-ViT-B/16 | 72.8 | 94.5 | 98.2 | 88.8 | 98.3 | 91.9 | 94.5 | 95.3 | 98.1 | 75.7 |
| | CLIP-ViT-L/14 | 75.9 | 95.7 | 99.2 | 93.0 | 99.1 | 94.8 | 95.3 | 95.4 | 98.3 | 80.5 |

Table 8: Performance of pre-trained and fine-tuned Flan-T5-base models on the 8-task NLP GLUE benchmark.

| Model | CoLA | MNLI | MRPC | QNLI | QQP | RTE | SST2 | STSB |
|---|---|---|---|---|---|---|---|---|
| Flan-T5-base (Pre-trained) | 69.1 | 56.5 | 76.2 | 88.4 | 82.1 | 80.1 | 91.2 | 62.2 |
| Flan-T5-base (Fine-tuned) | 75.0 | 83.4 | 87.5 | 91.5 | 85.4 | 85.9 | 93.6 | 88.7 |

## B.3 DETAILS OF BASELINES

In addition to the methods presented in Sec. 3.2, we introduce the following additional baselines.

- **Ties-Merging** An extension of Task Arithmetic that alleviates parameter redundancy and sign conflicts during model merging (Yadav et al., 2023). For task $t$, we first compute the task-specific difference vector $\tau_t = \theta_t - \theta_0$, which is then trimmed and sign-normalized. The update is defined as $\tau_t^{\text{Ties}} = \text{Ties}\left(\tau_{t-1}^{\text{Ties}}, \tau_t\right)$, and the merged model is updated by $\theta_t^{\text{merged}} = \theta_{t-1}^{\text{merged}} + \lambda\tau_t^{\text{Ties}}$.

- **Maximum Magnitude Selection (MAGMAX).** An extension of Task Arithmetic that selects, for each parameter dimension, the update with the larger absolute value (Marczak et al., 2024). Formally, $\tau_t^{\text{MagMax}} = \text{MagMax}(\tau_{t-1}^{\text{MagMax}}, \tau_t)$, and the merged model is updated as $\theta_t^{\text{merged}} = \theta_{t-1}^{\text{merged}} + \lambda\tau_t^{\text{MagMax}}$. In our experiments, we apply MAGMAX to merge individually fine-tuned models, denoted as MAGMAX-IND.

- **Weight-Ensembling MoE (WEMOE).** A mixture-of-experts based merging strategy (Tang et al., 2024b), where task-specific MLP layers serve as experts and are aggregated via a gating function. In the continual merging setting, however, WEMOE does not converge: the expert MLPs introduce excessive parameters, which makes learning the gating function unreliable with only a few unlabeled test samples. To mitigate this issue, we compress the MLP experts using LoRA, yielding the variant LORA-WEMOE. For test-time adaptation, we fine-tune the gating module using 5 randomly sampled instances per class from the test set of each new task.

- **WUDI-Merging.** A data-free merging method that reduces task interference by enforcing orthogonality between task vectors and their residual components (Cheng et al., 2025). In the

continual setting, each linear layer $l$ maintains two task vectors: the cumulative merged vector from previous tasks $\tau_{t-1}^{\text{merged},(l)} = \theta_{t-1}^{\text{merged},(l)} - \theta_0^{(l)}$ and the current task vector $\tau_t^{(l)} = \theta_t^{(l)} - \theta_0^{(l)}$. The merged vector $\tau_m^{(l)}$ is obtained by optimizing $\mathcal{L} = \sum_i \frac{1}{\|\tau_{i,l}\|_F^2} \left\| (\tau_{m,l} - \tau_{i,l})(\tau_{i,l})^\top \right\|_F^2$. The update follows gradient descent, and the final merged parameters are $\theta_t^{\text{merged}} = \theta_0 + \tau_m$.

**Details of Baseline Hyper-parameters.** As summarized in Tab. 9, we report the key hyperparameter settings for all baseline methods and task configurations. Notably, our approach employs a *single fixed* configuration applied uniformly across models (CLIP, Flan-T5), task scales (8, 14, 20), and all 10 task orders. This design highlights the robustness and generality of our method, ensuring that performance improvements do not stem from task-specific hyperparameter tuning.

Table 9: Hyperparameter settings for all baselines across different task configurations.

| Method | Tasks | Scale Factor ($\lambda$) | $\alpha$ | Top-k (%) | LR | Steps | $r_p$ | $r_l$ | $r_v$ |
|---|---|---|---|---|---|---|---|---|---|
| TASK ARITHMETIC | 8 | 0.3 | - | - | - | - | - | - | - |
| | 14/20 | 0.1 | - | - | - | - | - | - | - |
| TIES-MERGING | 8 | 0.3 | - | 20 | - | - | - | - | - |
| | 14/20 | 0.1 | - | 20 | - | - | - | - | - |
| MAGMAX-IND | 8/14/20 | 0.5 | - | - | - | - | - | - | - |
| LW. ADAMERGING | 8/14/20 | 0.3 | - | - | 1e-4 | 50 | - | - | - |
| LORA-WEMOE | 8/14/20 | 0.3 | - | - | 1e-4 | 50 | - | 64 | - |
| WUDI-MERGING | 8/14/20 | - | - | - | 1e-5 | 50 | - | - | - |
| OPCM | 8/14/20 | - | 0.5 | - | - | - | - | - | - |
| **NUFILT (Ours)** | 8/14/20 | - | - | - | 1e-3 | 50 | 128 | 64 | 8 |

### B.4 HYPER-PARAMETER ADJUSTMENT GUIDELINES

This section provides a brief guide for selecting the hyper-parameters used in NUFILT.

**Rank hyper-parameters** $(r_p, r_l, r_v)$. Our sensitivity analysis shows that all three rank hyper-parameters exhibit unimodal behavior and can be tuned independently with a small validation set:

- $r_p$: Controls how much of the previous-task subspace is preserved. Larger values reduce forgetting; moderate values work consistently across backbones.
- $r_l$: Low-rank correction term. Set smaller than $r_p$ to avoid disrupting the preserved subspace; small ranks are stable.
- $r_v$: Dimensionality of the task vector subspace. A small value ensures stable alignment; increasing it introduces weaker singular directions and typically does not improve performance.

**Number of Iterations.** Performance improves initially but saturates quickly. A single tuned value (e.g., 50–100 iterations) generalizes across different ViT backbones.

**Scaling Across Architectures.** Sensitivity patterns are consistent for ViT-B/32, ViT-B/16, and ViT-L/14. The same hyperparameter settings work well across model sizes, with only minimal fine-tuning needed when desired.

## C ADDITIONAL RESULTS

In this section, we provide additional experimental results to support the findings reported in the main paper. Specifically, we include: (1) detailed overall performance results (C.1); (2) extended visualizations on subspace alignment (C.2).

## C.1 DETAILED OVERALL PERFORMANCE RESULTS

Tab. 10 expands on the average results in Tab. 1 by reporting per-task average accuracy after continually merging 20 tasks. We compare six methods, WA, Task Arithmetic, Ties-Merging, MagMax-IND, OPCM, and our proposed NUFILT across three CLIP-ViT backbones (B/32, B/16, L/14). NUFILT achieves the highest accuracy on most tasks. These fine-grained results reinforce the main paper's findings, highlighting NUFILT's ability to improve performance on continual model merging.

## C.2 EXTENDED VISUALIZATIONS ON SUBSPACE ALIGNMENT

This section provides extended subspace alignment visualizations for three additional models (ViT-B/32, ViT-L/14, Flan-T5-base; see Fig. 6, 7, and 8) to further validate the universal alignment of task vectors with task-relevant representations across architectures and layers. All visualizations follow a unified 3×3 grid structure where columns represent model components (left: full model; middle: attention layers; right: MLP layers) and rows show visualization types (top: layer-wise mean affinity heatmaps; middle: layer-wise 90th percentile affinity heatmaps; bottom: layer-wise ECDF curves). The layer-wise ECDFs reveal consistent spectral overlaps: for ViT models (Fig. 6 and 7), high (MNIST Data →MNIST Vector), moderate (EuroSAT Data →RESISC45 Vector), and low (SVHN Data →DTD Vector); for Flan-T5-base (Fig. 8), high (RTE Data →RTE Vector), moderate (RTE Data →QNLI Vector), and low (RTE Data →SST-2 Vector). These results across both vision and NLP domains confirm the consistent alignment phenomenon originally observed in Fig. 2, with all affinity heatmaps showing strong diagonal dominance where matched pairs exhibit significantly higher affinity than mismatched ones.

# D DISCUSSIONS

## D.1 LIMITATIONS

Similar to prior approaches in model merging, our NUFILT framework is built on the assumption that all task-specific fine-tuned models $\{\theta_t\}_{t=1}^{T}$ originate from a common pre-trained initialization $\theta_0$. The implications of this assumption—such as its effect on the alignment of task vectors in the underlying subspace—remain insufficiently explored and merit further investigation. Our current experiments are restricted to models sharing the same backbone (*e.g.*, CLIP ViT variants, Flan-T5-base); extending the framework to heterogeneous initializations or architectures represents an interesting avenue for future work.

## D.2 BROADER IMPACTS

This paper aims to advance the Machine Learning field. Our work has potential societal impacts, but none require specific highlighting here.

## D.3 LLM USAGE

In preparing this submission, we used large language models (LLMs) solely as an assistive tool for sentence-level editing, including grammar correction, spelling adjustments, and minor word-choice refinements. The LLM was not involved in research ideation, methodological design, experimental analysis, or content generation beyond language editing. All substantive scientific contributions are solely those of the authors.

Table 10: Test set accuracy comparisons on different downstream tasks.

| | Model | SUN397 | Cars | RESISC45 | EuroSAT | SVHN | GTSRB | MNIST | DTD | Flowers102 | PCAM |
|---|---|---|---|---|---|---|---|---|---|---|---|
| ViT-B/32 | C. Fine-Tuned | 53.9 | 38.2 | 64.7 | 98.7 | 45.4 | 34.4 | 86.7 | 58.4 | 57.5 | 67.7 |
| | WA | 64.2 | 59.6 | 64.8 | 60.9 | 47.3 | 43.1 | 71.8 | 46.4 | 66.5 | 63.9 |
| | C.TA | 62.0 | 53.7 | 60.9 | 58.1 | 48.5 | 48.9 | 79.4 | 46.1 | 61.1 | 73.4 |
| | C.TIES | 62.5 | 49.1 | 55.8 | 50.9 | 54.6 | 49.3 | 82.0 | 46.7 | 58.5 | 69.9 |
| | MagMax-Ind | 63.6 | 53.1 | 59.7 | 49.1 | 53.8 | 53.1 | 79.8 | 43.2 | 56.9 | 75.1 |
| | LW AdaMerging | 63.1 | 60.0 | 63.5 | 60.1 | 35.6 | 32.1 | 51.8 | 45.4 | 66.6 | 60.2 |
| | LoRA-WEMOE | 51.4 | 45.8 | 63.3 | 43.5 | 42.9 | 34.6 | 58.9 | 46.5 | 47.5 | 60.1 |
| | WUDI-Merging | 59.4 | 51.6 | 63.8 | 63.2 | 63.6 | 61.9 | 87.7 | 48.1 | 53.5 | 72.1 |
| | OPCM | 64.4 | 51.1 | 66.0 | 71.7 | 66.1 | 56.0 | 90.2 | 40.4 | 64.9 | 80.2 |
| | **NUFILT (Ours)** | 60.3 | 53.6 | 67.0 | 76.6 | 87.4 | 85.6 | 98.1 | 51.8 | 55.5 | 82.4 |
| ViT-B/16 | C. Fine-Tuned | 62.7 | 58.0 | 67.6 | 99.1 | 46.0 | 29.2 | 93.9 | 61.9 | 64.1 | 75.2 |
| | WA | 67.1 | 64.6 | 69.3 | 63.4 | 62.4 | 52.7 | 80.7 | 46.6 | 71.8 | 63.1 |
| | TA | 65.8 | 57.5 | 63.8 | 59.5 | 64.7 | 54.0 | 88.0 | 45.3 | 67.5 | 67.1 |
| | TIES | 64.2 | 52.9 | 60.9 | 53.0 | 62.8 | 48.8 | 88.4 | 45.0 | 61.3 | 68.5 |
| | MagMax-Ind | 65.8 | 51.8 | 57.8 | 42.6 | 54.4 | 43.7 | 83.0 | 42.8 | 60.4 | 69.8 |
| | LW AdaMerging | 65.5 | 65.7 | 69.8 | 59.4 | 50.1 | 44.2 | 61.1 | 47.1 | 71.8 | 57.9 |
| | LoRA-WEMOE | 62.7 | 60.2 | 69.4 | 37.7 | 52.1 | 39.9 | 63.1 | 45.3 | 64.3 | 51.7 |
| | WUDI-Merging | 64.6 | 55.1 | 70.1 | 65.3 | 69.1 | 60.4 | 90.0 | 48.4 | 68.8 | 79.3 |
| | OPCM | 67.9 | 55.9 | 73.7 | 77.5 | 74.4 | 63.2 | 94.1 | 49.2 | 72.3 | 79.6 |
| | **NUFILT (Ours)** | 64.9 | 54.5 | 77.8 | 83.7 | 88.9 | 83.7 | 98.1 | 51.8 | 70.0 | 86.1 |
| ViT-L/14 | C. Fine-Tuned | 69.5 | 73.6 | 78.3 | 99.2 | 59.3 | 49.3 | 98.6 | 69.7 | 83.2 | 78.3 |
| | WA | 70.7 | 77.7 | 76.4 | 75.3 | 69.5 | 62.1 | 93.7 | 57.7 | 80.0 | 73.6 |
| | TA | 70.4 | 74.1 | 73.9 | 66.3 | 69.9 | 65.6 | 95.1 | 56.6 | 78.6 | 70.4 |
| | TIES | 69.7 | 70.3 | 65.3 | 47.9 | 76.1 | 63.6 | 94.7 | 54.4 | 77.9 | 72.3 |
| | MagMax-Ind | 73.1 | 73.7 | 75.6 | 64.6 | 73.7 | 68.8 | 94.6 | 56.1 | 78.0 | 71.7 |
| | LW AdaMerging | 68.8 | 78.6 | 75.9 | 65.7 | 58.3 | 51.6 | 79.9 | 57.4 | 80.6 | 52.4 |
| | LoRA-WEMOE | 62.1 | 68.1 | 68.7 | 53.2 | 47.5 | 49.4 | 69.8 | 49.1 | 66.2 | 54.2 |
| | WUDI-Merging | 74.1 | 88.0 | 83.8 | 77.0 | 78.2 | 79.7 | 95.8 | 63.7 | 91.4 | 80.1 |
| | OPCM | 73.1 | 78.3 | 82.4 | 80.2 | 80.8 | 80.4 | 97.4 | 61.6 | 84.8 | 76.3 |
| | **NUFILT (Ours)** | 74.8 | 82.3 | 87.6 | 90.1 | 93.6 | 94.4 | 98.6 | 66.2 | 94.5 | 86.4 |

| | Model | FER2013 | OxfordIIITPet | STL10 | CIFAR100 | CIFAR10 | Food101 | FashionMNIST | EMNIST | KMNIST | RenderedSST2 |
|---|---|---|---|---|---|---|---|---|---|---|---|
| ViT-B/32 | C. Fine-Tuned | 58.3 | 68.5 | 86.7 | 40.2 | 70.5 | 50.0 | 90.7 | 72.4 | 54.5 | 54.5 |
| | WA | 50.2 | 84.1 | 97.0 | 69.8 | 92.7 | 80.4 | 71.3 | 15.0 | 11.5 | 61.8 |
| | TA | 51.4 | 82.3 | 94.9 | 64.6 | 91.4 | 71.9 | 73.9 | 17.8 | 12.2 | 59.9 |
| | TIES | 49.5 | 81.3 | 95.2 | 63.7 | 91.2 | 70.2 | 73.7 | 17.8 | 16.9 | 59.8 |
| | MagMax-Ind | 56.5 | 79.9 | 94.6 | 68.7 | 91.9 | 73.8 | 74.3 | 18.3 | 15.4 | 63.9 |
| | LW AdaMerging | 43.2 | 83.7 | 96.8 | 67.0 | 89.9 | 81.6 | 63.7 | 16.8 | 10.7 | 59.1 |
| | LoRA-WEMOE | 44.6 | 72.5 | 86.1 | 40.1 | 63.8 | 63.8 | 48.1 | 10.3 | 12.8 | 55.7 |
| | WUDI-Merging | 60.7 | 80.3 | 93.1 | 60.9 | 85.4 | 62.8 | 76.1 | 38.2 | 21.9 | 70.2 |
| | OPCM | 55.8 | 82.9 | 95.9 | 67.6 | 92.8 | 74.0 | 76.3 | 22.4 | 18.3 | 64.6 |
| | **NUFILT (Ours)** | 62.5 | 81.6 | 95.1 | 63.1 | 91.1 | 66.0 | 85.4 | 42.6 | 43.5 | 71.6 |
| ViT-B/16 | C. Fine-Tuned | 60.5 | 84.5 | 90.5 | 38.8 | 73.6 | 61.9 | 89.7 | 83.3 | 51.5 | 72.8 |
| | WA | 50.9 | 89.6 | 98.0 | 72.9 | 94.2 | 85.9 | 73.3 | 15.6 | 12.4 | 62.5 |
| | TA | 50.7 | 89.3 | 97.0 | 68.0 | 93.1 | 80.3 | 75.7 | 18.1 | 16.7 | 61.8 |
| | TIES | 50.4 | 87.9 | 96.3 | 63.1 | 91.7 | 78.0 | 75.0 | 23.4 | 24.9 | 61.5 |
| | MagMax-Ind | 57.7 | 88.8 | 97.5 | 71.5 | 94.4 | 81.3 | 77.2 | 24.5 | 25.0 | 59.4 |
| | LW AdaMerging | 46.8 | 88.9 | 98.1 | 69.2 | 91.4 | 86.6 | 67.2 | 17.2 | 11.0 | 59.2 |
| | LoRA-WEMOE | 45.6 | 91.2 | 92.3 | 41.3 | 64.3 | 78.1 | 48.0 | 23.5 | 16.6 | 52.7 |
| | WUDI-Merging | 64.7 | 91.5 | 95.9 | 67.5 | 90.2 | 78.3 | 81.4 | 50.8 | 30.5 | 70.0 |
| | OPCM | 59.5 | 91.8 | 97.7 | 73.2 | 94.7 | 83.1 | 81.3 | 26.5 | 23.4 | 66.8 |
| | **NUFILT (Ours)** | 66.0 | 92.3 | 97.0 | 70.5 | 94.0 | 80.8 | 88.3 | 69.9 | 71.0 | 72.1 |
| ViT-L/14 | C. Fine-Tuned | 68.0 | 92.1 | 94.5 | 60.5 | 85.7 | 74.8 | 93.1 | 89.0 | 59.2 | 78.8 |
| | WA | 52.7 | 94.2 | 99.2 | 81.7 | 97.0 | 90.7 | 77.4 | 16.1 | 10.4 | 66.1 |
| | TA | 55.7 | 94.2 | 98.6 | 79.1 | 91.6 | 87.6 | 80.8 | 17.6 | 10.6 | 63.6 |
| | TIES | 57.6 | 93.5 | 97.8 | 74.0 | 95.6 | 84.7 | 79.7 | 20.2 | 12.6 | 58.4 |
| | MagMax-Ind | 52.9 | 93.9 | 98.7 | 82.1 | 97.3 | 89.5 | 81.6 | 19.2 | 11.1 | 68.4 |
| | LW AdaMerging | 49.2 | 93.5 | 99.3 | 77.2 | 95.8 | 91.1 | 68.2 | 18.6 | 9.8 | 66.6 |
| | LoRA-WEMOE | 46.3 | 84.5 | 87.6 | 52.1 | 70.5 | 73.3 | 50.0 | 18.7 | 10.9 | 56.5 |
| | WUDI-Merging | 66.2 | 95.6 | 98.6 | 79.5 | 95.5 | 99.1 | 84.0 | 46.1 | 23.9 | 78.3 |
| | OPCM | 61.8 | 95.4 | 99.2 | 83.0 | 97.8 | 90.9 | 86.0 | 26.4 | 14.7 | 71.0 |
| | **NUFILT (Ours)** | 67.0 | 95.8 | 98.8 | 79.6 | 96.5 | 90.4 | 91.9 | 51.0 | 74.3 | 77.2 |

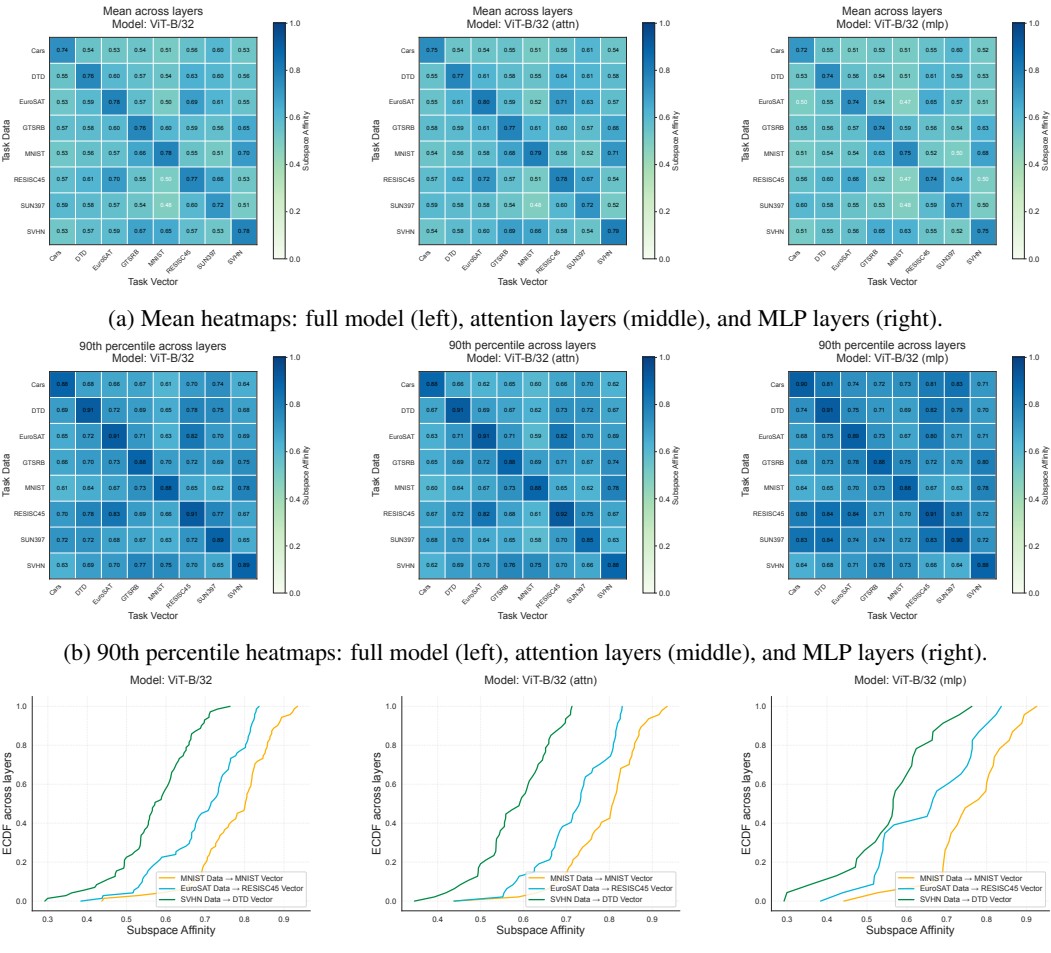

(a) Mean heatmaps: full model (left), attention layers (middle), and MLP layers (right).

(b) 90th percentile heatmaps: full model (left), attention layers (middle), and MLP layers (right).

(c) ECDF distributions: full model (left), attention layers (middle), and MLP layers (right).

Figure 6: Subspace affinity between data and task vectors in ViT-B/32 across eight datasets.

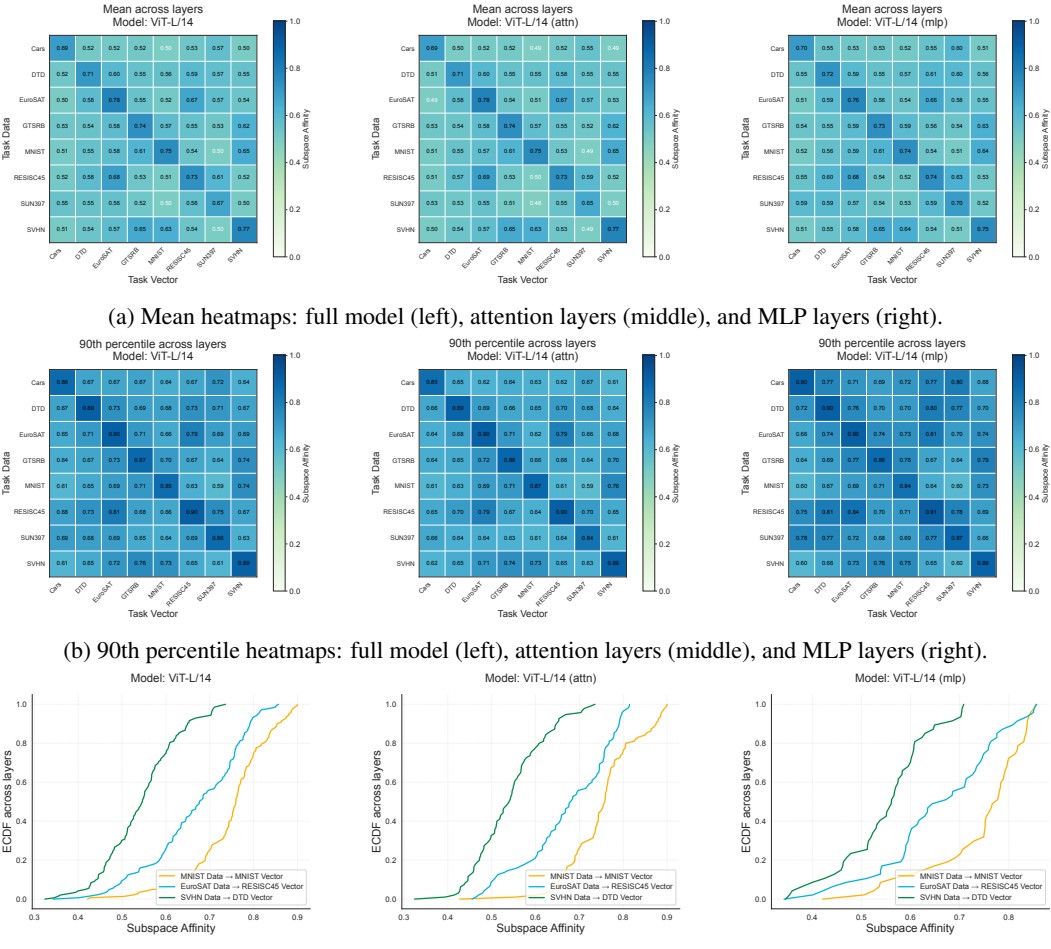

(a) Mean heatmaps: full model (left), attention layers (middle), and MLP layers (right).

(b) 90th percentile heatmaps: full model (left), attention layers (middle), and MLP layers (right).

(c) ECDF distributions: full model (left), attention layers (middle), and MLP layers (right).

Figure 7: Subspace affinity between data and task vectors in ViT-L/14 across eight datasets.

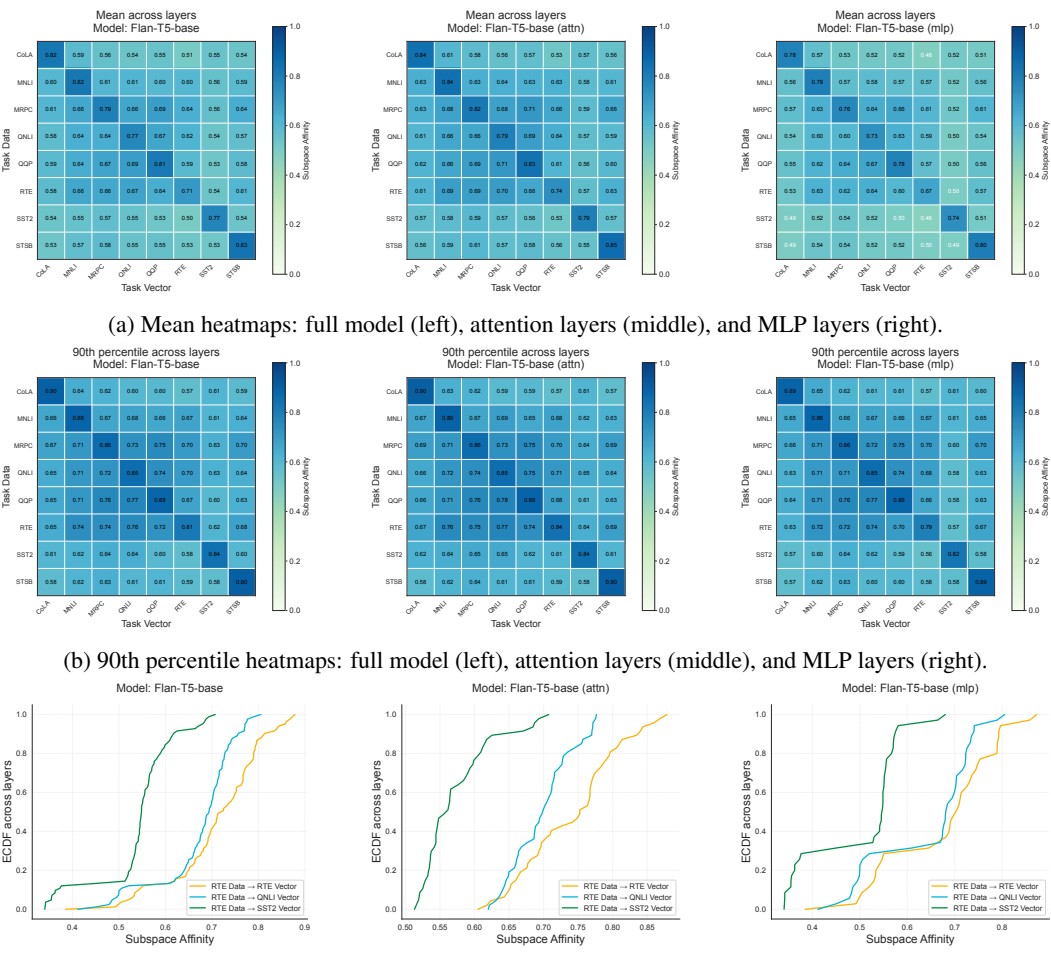

(a) Mean heatmaps: full model (left), attention layers (middle), and MLP layers (right).

(b) 90th percentile heatmaps: full model (left), attention layers (middle), and MLP layers (right).

(c) ECDF distributions: full model (left), attention layers (middle), and MLP layers (right).

Figure 8: Subspace affinity between data and task vectors in Flan-T5-base across eight datasets.

