# OpenReview forum: "Null-Space Filtering for Data-Free Continual Model Merging: Preserving Stability, Promoting Plasticity"
_ICLR.cc/2026/Conference — ICLR 2026 Poster_

### Official Review · Reviewer_m3Kf · 2025-10-28

**Soundness:** 3
**Presentation:** 3
**Contribution:** 3
**Rating:** 6
**Confidence:** 4

**Summary:**

This paper proposes Null-space Filtering to address the data-free continual model merging problem: sequentially merging multiple fine-tuned models into a single backbone to cover all task label spaces, without the original training data and only with access to the current task model and the merged backbone network. The authors formalize the problem as two data-level desiderata: transparency and fidelity, and propose a strategy to map these data-level objectives into parameter space: based on the observation that the main subspace of the task vector is approximately aligned with the representation subspace. The paper provides detailed vision and NLP experiments showing that NUFILT outperforms multiple recent baseline methods in ACC and BWT metrics.

**Strengths:**

1. The writing is quite easy to read and it was well-written
2. The idea of ​​constructing a null-space projection based on empirical findings is both intuitive and mathematically connected, providing a reasonable path for "replacing data supervision with parametric geometric structures in the absence of data."
3. The paper studies multiple base models
4. The gains from their proposed method of sampling are convincing and comprehensive

**Weaknesses:**

1. The use of Null Space has been applied in many reccent works, such as [1] [2] [3], so the contribution of the work is incremental. The author should further discuss the relation between this works.
2. The paper uses quantities such as singular vectors representing the data, the largest singular value of a matrix, and σ1(X) to derive bounds (e.g., Corollary 1). However, in the strict case of data obscurity, how these quantities are estimated/upper bounded in practice needs to be more clearly explained. The current bridge between theoretical assumptions and practical implementations is not transparent enough.
3. The paper claims to use a set of global hyperparameters, but does not provide guidance on hyperparameter adjustment for different backbones or longer sequences; in addition, it is unclear how the number of training iterations, rank selection, etc. scale to different model sizes.




[1] Alphaedit: Null-space constrained knowledge editing for language models

[2] Multi-task model merging via adaptive weight disentanglement

[3] Orthogonal Subspace Learning for Language Model Continual Learning

**Questions:**

1. What are the results of NUFILT when apply to larger scale models like qwen and llama ?
2. Wiil the initialization of LoRA affect the merging result。

If the author can solve the question and the weakness well, i will raise my score.

---

> ### Author Response · Authors · 2025-11-24
> **Respond to Reviewer m3Kf (1/n)**
>
> # W1 — Relation between prior works
>
> Thank you for your valuable feedback. We clarify the distinctions between our method and prior orthogonality-based approaches below.
>
> **(1) Difference from [1].**
>  [1] projects parameter updates into the null space of a knowledge base, but the delta parameters must be obtained through task-specific training data. Our method is fully data-free: we extract task vectors directly from the pretrained models and estimate the null space using the right singular vectors of previously merged task vectors, requiring no data at any stage. After projection, we apply a lightweight LoRA-based optimization to improve adaptation.
>
> **(2) Difference from multi-task merging in [2].**
>  [2] assumes access to all task vectors simultaneously and optimizes a joint solution for *multi-task merging.* Our method instead addresses *continual model merging*, operating sequentially with minimal overhead and requiring only the previously merged model and the current task vector.
>
> **(3) Difference from LoRA-based orthogonality in [3].**
>  [3] projects new LoRA parameters into the null space of old LoRA parameters in a data-driven continual learning setting. We project new task vectors into the null space of *previous task representations*, estimated from merged task vectors without using any data. This yields a fully data-free continual merging framework.
>
> For clarity, we have added a concise comparison of these orthogonality-based approaches in the related work section.
>
> # W2 — Data representing quantities
>
> Thank you for the question. The quantities such as singular vectors of the data matrix and the spectral term $ \sigma _ {1}(X)$ appear only in the theoretical analysis to establish upper bounds on subspace alignment. These terms are not estimated in practice. Instead, the theory motivates the structure of our method—specifically, that projecting onto the dominant subspace of previous tasks is sufficient to control interference.
>
> In the practical algorithm, we use three hyperparameters $(r _ p, r _ l, r _ v)$ to specify the dimensionalities of the retained or removed subspaces, rather than computing $\sigma _ {1}(X)$ or data singular vectors directly. These hyperparameters serve as model-level controls and can be set globally across tasks. As shown in our experiments, a single configuration generalizes well across datasets and backbones, and no data-driven estimation of the theoretical quantities is required. The theoretical results thus provide intuition about why low-rank subspace selection is effective, while the practical implementation replaces the unobservable data-dependent terms with fixed hyperparameter choices.
>
> # W3 — Guidance on hyperparameter adjustment
>
> Thank you for the insightful question. We now clarify how the hyperparameters behave across architectures and model sizes. In Sec. *Hyper-parameter Sensitivity*, we analyze the three rank hyperparameters $(r _ p, r _ l, r _ v)$ and their empirical effects.
>
> Our experiments consistently show that these hyperparameters follow unimodal sensitivity patterns, and thus can be adjusted independently using a small validation set. In practice:
>
> -  **$r _ p$** controls how much of the previous-task subspace is preserved. Larger values offer stronger forgetting prevention and remain stable across backbones.
> -  **$r _ l$** is kept smaller than $r _ p$ to avoid disrupting the null-space structure, ensuring that only a limited number of directions are updated.
> -  **$r _ v$** governs the size of the task-vector subspace and is generally chosen small for stable alignment.
>
> Regarding **iterations**, Table 5 shows that performance gains saturate quickly: increasing iterations improves accuracy at first but plateaus after a relatively small number of steps. The iteration count can therefore be tuned once on a validation set and reused across backbones.
>
> In terms of scaling to different model sizes, the sensitivity to hyperparameters generally remains consistent, and the same tuning procedure can be applied. We have observed that the same empirical settings work well across various backbone architectures and task sizes, though fine-tuning based on the specific task or model may still yield further improvements.
>
> For completeness, we have added a short hyperparameter adjustment guide in the appendix.

---

> ### Author Response · Authors · 2025-11-24
> **Respond to Reviewer m3Kf (2/2)**
>
> # Q1 — Scalability to larger models
>
> Thank you for raising this point. To address the scalability of our approach to larger models, we have extended our evaluation to include **LLaVA-1.5-7B**.
>
> Our experiments demonstrate that the proposed framework is capable of efficiently scaling to larger models, confirming the robustness and generalizability of our approach. Notably, our method is model-agnostic. We have previously validated it on CLIP (contrastive architecture) and Flan-T5 (encoder-decoder architecture), and now, we have extended it to LLaVA-1.5-7B (decoder-only architecture), further showcasing its versatility. We have included these results in the revised version of the paper. (see section 5.2)
>
> Table M.1: Results of continual merging LLaVA models on 4 tasks, ordered alphabetically.
>
> | Method       | IconQA | Image | SciQA | VizWiz | Average |
> | ------------ | ------ | ----- | ----- | ------ | ------- |
> | Individual   | 75.5   | 96.0  | 83.7  | 64.8   | 80.0    |
> | Pre-Trained  | 14.1   | 40.9  | 61.7  | 41.0   | 39.4    |
> | TA           | 47.6   | 67.9  | 74.2  | 46.6   | 59.1    |
> | TIES-MERGING | 69.6   | 74.4  | 75.5  | 44.6   | 66.0    |
> | OPCM         | 58.0   | 86.4  | 81.1  | 44.9   | 67.6    |
> | NUFILT       | 59.6   | 93.6  | 80.6  | 48.3   | 70.5    |
>
> # Q2 — LoRA initialization
>
> Thank you for the insightful question. In our method, LoRA modules serve as *lightweight adaptive refinements* inside the null-space filter, and we follow the standard LoRA initialization, where A is randomly initialized with a small Gaussian distribution and B is initialized to zero. Therefore, the LoRA initialization does not alter the null-space projection, ensuring that the transparency constraint is fully preserved at the start of optimization.
>
> We conducted additional experiments with several alternative initialization schemes (e.g., Xavier, Kaiming, both A/B non-zero). Results show that NUFILT is highly robust to LoRA initialization, and all variants converge to similar performance.
>
> Table M.2: Effect of different LoRA initialization (CLIP ViT-B/32, 8 tasks)
>
> | Init ID                         | A Initialization | B Initialization | ACC (%)  | BWT (%)  |
> | ------------------------------- | ---------------- | ---------------- | -------- | -------- |
> | Init-0 (Default, used in paper) | 𝒩(0, 0.02²)      | 0                | 83.6±0.2 | −2.7±0.7 |
> | Init-1                          | Xavier Uniform   | 0                | 83.4±0.3 | −2.9±0.8 |
> | Init-2                          | Xavier Normal    | 0                | 83.4±0.3 | −2.9±0.8 |
> | Init-3                          | 𝒩(0, 0.02²)      | 𝒩(0, 0.02²)      | 83.5±0.3 | −2.7±0.8 |
> | Init-4                          | 𝒩(0, 0.01²)      | 𝒩(0, 0.01²)      | 83.6±0.3 | −2.6±0.8 |

---

> ### Comment · Reviewer_m3Kf · 2025-11-24
> **Reply to author's rebuttal**
>
> the author's rebuttal address most of my concerns, i have raised my score accordingly.

---

> > ### Author Response · Authors · 2025-11-24
> >
> > Thank you for the helpful comments and for reconsidering your evaluation. Your feedback has been valuable for improving the paper.

---

### Official Review · Reviewer_GLxk · 2025-10-29

**Soundness:** 4
**Presentation:** 4
**Contribution:** 3
**Rating:** 8
**Confidence:** 5

**Summary:**

This paper proposes NUFILT for data-free continual model merging (DFCMM). Each new task’s update (task vector) is first projected onto the null space of a low-rank subspace spanned by the previously merged updates. Then a lightweight LoRA term is trained (using a projection-based surrogate objective) to recover task-specific signal. Finally, the projection and LoRA are fused back into the backbone so no extra parameters remain at inference. The paper motivates the projection with an “approximate alignment” claim between task-vector subspaces and data-representation subspaces, and reports gains over WA/TA, OPCM, WUDI-Merging and others on CLIP vision suites and GLUE with Flan-T5.

**Strengths:**

- The paper present a simple, practical recipe. The three-step pipeline is easy to follow.
- The proposed method elegantly combines null-space projection with parameter-efficient LoRA adaptation, supported by solid mathematical reasoning and empirical evidence of subspace alignment. The approach is conceptually simple yet technically sound.
- Experiments are thorough and well-controlled, covering vision and language tasks, multiple architectures, and extensive baselines. The ablations and sensitivity analyses are carefully executed, providing convincing evidence of robustness and practical usability.

**Weaknesses:**

I overall like this paper, which is clearly written and technically rigorous. I have only few questions.

1. In Fig. 4(a), the accuracy drops when introducing a small projection rank
(r_p = 1-4), even lower than using no projection (r_p = 0). Could the authors clarify why a small null-space rank initially hurts performance? Additionally, backward transfer first decreases and then improves as r_p increases, what causes this non-monotonic behavior?

2. The subspace affinity is defined as $A(V_d,\hat V)=\frac1{r_d}\|\hat V^\top V_d\|_F^2\in[0,1]$ , but Theorem 1 defines $A(V_d,\hat V)=1-\frac1{r_d}\|\hat V^\top V_d\|_F^2$ and claims $A\le\zeta^2$ (Eq. 6). I might have misunderstood. Please check this.

3. The method repeatedly requires SVDs per layer on $\tilde\tau_{\le t-1}$ and on $\tau_t$  but runtime in Table 4 only reports the adaptation loop, not the SVD cost.

4. Although the robustness analysis is informative, the chosen global ranks might still require adjustment for different model scales. A short discussion on adaptive or automatic rank selection would strengthen the paper’s practical perspective.

**Questions:**

see weekness

---

> ### Author Response · Authors · 2025-11-24
> **Respond to Reviewer GLxk (1/n)**
>
> # W1 — Explaination to non-monotonic behavior
>
> Thank you for the question. The non-monotonic pattern in Fig. 4(a) is expected and is related to how many **important directions** (dominant singular directions) the previous tasks occupy.
>
> When the projection rank is small ($r _ p = 1$ – $4$), only a **portion** of these directions is removed. In this example, the old-task subspace clearly has more than four meaningful directions. Partially removing the subspace while leaving other important directions intact **breaks its structural coherence**. Since the data typically depends on several dominant directions jointly, this partial filtering can hurt performance—sometimes even more than applying no projection at all ($r _ p = 0$).
>
> This also explains the non-monotonic behavior of backward transfer. Small $r_p$ values cause stronger forgetting because only part of the old-task subspace is filtered. Once $r _ p$ becomes large enough to cover the intrinsic dimensionality of the old tasks (e.g., $r _ p \ge 4$ in this case), the projection removes **almost the entire** old-task subspace. Past this point, increasing $r _ p$ leads to a **more consistent and monotonic** improvement in BWT.
>
> In practice, we recommend using a larger $r _ p$, which removes all major old-task directions and avoids the unstable regime caused by partial subspace filtering.
>
>
> # W2 — Clarification on Subspace Affinity Notation
>
> Thank you for the helpful comment. The issue comes from a **notation overload** between the main text and Theorem 1. In the main text, we define the subspace affinity as  $\mathcal{A}(V_d,\hat V) = \tfrac{1}{r_d}\|\hat V^\top V_d\|_F^2$.  However, in Theorem 1 we unintentionally used the *complementary* quantity  $1 - \tfrac{1}{r_d}\|\hat V^\top V_d\|_F^2$  but still denoted it with the same symbol, which caused the confusion.
>
> To resolve this overload, we revised Theorem 1 to explicitly state  $1 - \mathcal{A}(V_d^{(l)}, \hat V^{(l)}) \le \zeta^2$,  where  $\mathcal{A}(V_d^{(l)}, \hat V^{(l)}) = \tfrac{1}{r_d}\|\hat V^\top V_d\|_F^2$.
>
> This ensures that the meaning of $\mathcal{A}$ is **fully consistent** throughout the paper.  All changes are highlighted in blue in the revised PDF.
>
> # W3 — Computational cost of SVD
>
> Thank you for pointing this out. We have clarified the cost of the SVD step in our updated results. In Table G.1, the total runtime (“Tot.”) includes both the solving time of the optimization loop (“Solv.”) and the SVD computations. Therefore, the SVD overhead can be directly obtained as  **SVD cost = Tot. − Solv.**
> For example, with 50 iterations, NUFILT has a solving time of 18.4s and a total time of 138.9s, implying that approximately 120.5s are spent on SVDs across layers.
>
> Compared with baselines, training-free methods (TA/OPCM) remain the most efficient. Data-dependent methods (e.g., LW AdaMerging, LoRA-WEMOE) incur the highest cost due to multi-step training and data loading. Our method sits between the two extremes: its solving time is low due to the lightweight LoRA-based formulation, while the total time is moderately higher because of the additional SVD step. Overall, NUFILT offers the strongest accuracy while maintaining competitive efficiency among data-free merging methods.
>
>
>
> Table G.1: Computation Overhead
>
> | Model             | Iters. | Solv. | Tot.   | GPU Mem. | Param. | ACC (%)    |
> | ----------------- | ------ | ----- | ------ | -------- | ------ | ---------- |
> | **TA**            | –      | –     | 8.3s   | –        | 87.5M  | 67.5 ± 0.0 |
> | **OPCM**          | –      | –     | 76.6s  | 1.3GB    | 87.5M  | 75.5 ± 0.5 |
> | **LW AdaMerging** | 50     | 47.1s | 724.2s | 1.6GB    | 87.5M  | 53.4 ± 3.2 |
> | **LoRA-WEMOE**    | 50     | 72.9s | 754.6s | 1.8GB    | 103.7M | 68.8 ± 7.8 |
> | **WUDI-Merging**  | 50     | 28.9s | 37.7s  | 2.0GB    | 87.5M  | 74.7 ± 6.6 |
> | **NUFILT (ours)** | 25     | 9.3s  | 129.8s | 1.8GB    | 87.5M  | 83.3 ± 0.3 |
> | **NUFILT (ours)** | 100    | 35.9s | 156.4s | 1.8GB    | 87.5M  | 83.6 ± 0.2 |
> | **NUFILT (ours)** | 50     | 18.4s | 138.9s | 1.8GB    | 87.5M  | 83.6 ± 0.2 |

---

> ### Author Response · Authors · 2025-11-24
> **Respond to Reviewer GLxk (2/2)**
>
> # W4 — Guidance on hyperparameter adjustment
>
> Thank you for the insightful question. We now clarify how the hyperparameters behave across architectures and model sizes. In Sec. *Hyper-parameter Sensitivity*, we analyze the three rank hyperparameters $(r_p, r_l, r_v)$ and their empirical effects.
>
> Our experiments consistently show that these hyperparameters follow unimodal sensitivity patterns, and thus can be adjusted independently using a small validation set. In practice:
>
> -  **$r_p$** controls how much of the previous-task subspace is preserved. Larger values offer stronger forgetting prevention and remain stable across backbones.
> -  **$r_l$** is kept smaller than $r_p$ to avoid disrupting the null-space structure, ensuring that only a limited number of directions are updated.
> -  **$r_v$** governs the size of the task-vector subspace and is generally chosen small for stable alignment.
>
> Regarding **iterations**, Table 5 shows that performance gains saturate quickly: increasing iterations improves accuracy at first but plateaus after a relatively small number of steps. The iteration count can therefore be tuned once on a validation set and reused across backbones.
>
> In terms of scaling to different model sizes, the sensitivity to hyperparameters generally remains consistent, and the same tuning procedure can be applied. We have observed that the same empirical settings work well across various backbone architectures and task sizes, though fine-tuning based on the specific task or model may still yield further improvements.
>
> For completeness, we have added a short hyperparameter adjustment guide in the appendix.

---

### Official Review · Reviewer_SpsW · 2025-11-01

**Soundness:** 2
**Presentation:** 2
**Contribution:** 3
**Rating:** 4
**Confidence:** 4

**Summary:**

This paper proposes NUFILT (NUllspace FILTering), a lightweight, data-free adaptation framework for Data-Free Continual Model Merging (DFCMM) aimed at improving transparency and fidelity. The authors argue that task vectors align with data representation subspaces and provide theoretical guarantees for approximate subspace alignment. Empirically, NUFILT reports 4–7% gains in average accuracy over prior work, with minimal backward transfer (i.e., limited forgetting of previously learned tasks), and requires only 50 optimisation steps.

**Strengths:**

(1) Leveraging null-space filtering for data-free continual model merging is an interesting novel idea.

(2) The adaptation strategy is lightweight and inexpensive, making it broadly applicable in data-restricted settings.

(3) Empirical results validate that NUFILT improves average accuracy while maintaining low forgetting.

**Weaknesses:**

(1) The paper presents transparency and fidelity as distinct desiderata, but they are closely related and longstanding goals in continual learning. As such, their framing does not feel like a novel contribution.

(2) Despite emphasizing transparency and fidelity, the paper does not quantitatively track or compare these properties for NUFILT vs. baselines.

(3) Evaluation is limited to relatively older architectures (CLIP and T5). Results on modern LLMs would materially strengthen the paper’s relevance.

(4) The work also lacks comparison against more recent merging techniques, such as ISO-Merging[1], KnOTS[2], TSV-M[3] which also use the SVD decomposition to align task vectors.

(5) Notation and clarity issues:

(a) Terms are introduced without definition (e.g., “representation covariance” at L214; use of $\sigma$ at L249).

(b) Overloaded symbols (e.g., T in Algorithm 1).

(c) Non-standard or unclear formulations (e.g., Eq. (3–4): the product of X with task vectors is not well specified).

(d) Undefined acronyms (e.g., ECDF).

Collectively, these hinder readability and reproducibility.

References:

[1] Marczak, D., Magistri, S., Cygert, S., Twardowski, B., Bagdanov, A. D., & van de Weijer, J. (2025). No task left behind: Isotropic model merging with common and task-specific subspaces. arXiv preprint arXiv:2502.04959.

[2] Stoica, G., Ramesh, P., Ecsedi, B., Choshen, L., & Hoffman, J. (2024). Model merging with svd to tie the knots. arXiv preprint arXiv:2410.19735.

[3] Gargiulo, A. A., Crisostomi, D., Bucarelli, M. S., Scardapane, S., Silvestri, F., & Rodola, E. (2025). Task singular vectors: Reducing task interference in model merging. In Proceedings of the Computer Vision and Pattern Recognition Conference (pp. 18695-18705).

**Questions:**

(1) Hyperparameter sensitivity: Fig. 4 suggests sensitivity to $r_p, r_l, r_v$. In a data-free setting, how were these tuned? Is there a data-free or proxy procedure (e.g., curvature/NTK proxies, spectral heuristics, or stability criteria) to select them?

(2) Rank assumption: L296 assumes $r_v \geq r_d$, yet the implementation fixes it to a very low value $r_v = 8$ which may not guarantee the above assumption is met. Does this have any implications?

**Details Of Ethics Concerns:**

No concerns.

---

> ### Author Response · Authors · 2025-11-24
> **Respond to Reviewer SpsW (1/n)**
>
> # W1 — Introduced new terminologies
>
> Thank you for the comment. Stability and plasticity are indeed broad and widely used notions in continual learning. In our work, *transparency* and *fidelity* refer to **their concrete instantiations in the setting of continual model merging**, where the basic operations involve combining task vectors rather than updating models through data.
>
> Specifically, continual model merging requires reasoning about how task vectors interact in parameter space. Transparency and fidelity therefore capture the two key aspects of this interaction:
>
> -  **Transparency** formalizes how the new task vector affects the parameter responses associated with previously merged tasks.
> -  **Fidelity** formalizes how well the merged parameters preserve the contribution of the new task vector.
>
> These notions are thus aligned with classical continual learning goals, but expressed at the **parameter level** that is unique to model merging. We have updated the text to clarify this relationship and avoid potential ambiguity.
>
> We have updated Section 3.3 to more clearly articulate how these concepts relate to classical stability-plasticity concepts, and **removed** the statement about transparency and fidelity being main contributions in the introduction.
>
> # W2 — Quantitative evaluation of transparency and fidelity
>
> Thank you for the helpful suggestion. We have added quantitative measurements of both transparency and fidelity for NUFILT and baselines. Using the definitions in Eqs. (3) and (4), we compute layer-wise deviations on previously learned tasks (transparency) and deviations from the individual model of the current task (fidelity).
>
> As shown in the newly added Figure 5 (Section 5.3), NUFILT achieves substantially lower transparency loss across tasks, indicating minimal disturbance to earlier task representations. Its fidelity loss are also consistently lower, demonstrating that the merged parameters remain close to the individual solution w.r.t layer activiations. We additionally report the harmonic mean of the two metrics, which further highlights that NUFILT provides the best overall balance between  the two criteria.
>
> These results have been included in the revised paper.
>
> # W3 — Scalability to larger models
>
> Thank you for raising this point. To address the scalability of our approach to larger models, we have extended our evaluation to include **LLaVA-1.5-7B**.
>
> Our experiments demonstrate that the proposed framework is capable of efficiently scaling to larger models, confirming the robustness and generalizability of our approach. Notably, our method is model-agnostic. We have previously validated it on CLIP (contrastive architecture) and Flan-T5 (encoder-decoder architecture), and now, we have extended it to LLaVA-1.5 (decoder-only architecture), further showcasing its versatility. We have included these results in the revised version of the paper. (see section 5.2)
>
> Table S.1: Results of continual merging LLaVA models on 4 tasks, ordered alphabetically.
>
> | Method       | IconQA | Image | SciQA | VizWiz | Average |
> | ------------ | ------ | ----- | ----- | ------ | ------- |
> | Individual   | 75.5   | 96.0  | 83.7  | 64.8   | 80.0    |
> | Zero-Shot    | 14.1   | 40.9  | 61.7  | 41.0   | 39.4    |
> | TA           | 47.6   | 67.9  | 74.2  | 46.6   | 59.1    |
> | TIES-MERGING | 69.6   | 74.4  | 75.5  | 44.6   | 66.0    |
> | OPCM         | 58.0   | 86.4  | 81.1  | 44.9   | 67.6    |
> | NUFILT       | 59.6   | 93.6  | 80.6  | 48.3   | 70.5    |
>
> # W4 — Comparison to more recent  methods
>
> Thank you for the suggestion. We have added comparisons with more recent SVD-based merging methods, including ISO-Merging, KnOTS, and TSV-M. These methods are evaluated under the same continual merging setting with ViT backbones of different scales (ViT-B/32, ViT-B/16, and ViT-L/14). As shown in Table S.2, our method consistently achieves higher average accuracy across tasks and model sizes. We will include these results in Table 1 of the revised version.
>
>
>
> Table S.2: comparison to more recent  methods.
>
> | Method     |          | **ViT-B/32** |          |          | **ViT-B/16** |          |          | **ViT-L/14** |          |
> | ---------- | -------- | ------------ | -------- | -------- | ------------ | -------- | -------- | ------------ | -------- |
> |            | 8 tasks  | 14 tasks     | 20 tasks | 8 tasks  | 14 tasks     | 20 tasks | 8 tasks  | 14 tasks     | 20 tasks |
> | Iso-C      | 71.7±1.2 | 73.2±1.8     | 67.6±0.8 | 78.5±1.2 | 79.7±1.3     | 73.0±1.1 | 86.9±0.5 | 86.9±1.8     | 80.9±0.8 |
> | KnOTS-TIES | 54.4±6.9 | 67.8±0.4     | 60.5±1.4 | 57.9±8.4 | 71.6±0.3     | 62.7±1.1 | 68.3±5.7 | 78.8±0.3     | 69.7±0.8 |
> | TSV-M      | 68.2±4.8 | 63.3±4.8     | 58.8±3.3 | 75.4±4.0 | 69.2±3.1     | 63.1±1.3 | 82.2±3.6 | 78.1±3.6     | 70.5±1.2 |
> | NUFILT     | 83.6±0.2 | 78.0±0.2     | 71.0±0.9 | 87.3±0.1 | 83.1±0.3     | 78.1±0.9 | 91.6±0.1 | 89.2±0.1     | 84.7±0.8 |

---

> ### Author Response · Authors · 2025-11-24
> **Respond to Reviewer SpsW (2/n)**
>
> # W5 — Notation and clarity issues
>
> ## (a) representation covariance” at L214; use of $\sigma$ at L249
>
> Thank you for your comment. We have now updated the manuscript to provide clear definitions for all terms that were previously introduced without explanation. Specifically: (1) The term representation covariance (L213) has been defined explicitly in the text, along with its mathematical formulation using the uncentered covariance; (2) We have clarified the definition of $\sigma_{r_d}(T_0)$ (L241), which denotes the $r_d$-th largest singular value of the matrix $T_0$.
>
> ## (b) Overloaded symbols
>
> Thank you for your careful review. We have revised the notation to avoid overloaded symbols. Specifically, the symbol $T$ used in the Equation 17 has been replaced with $\mathcal{T}$ ensuring that it does not conflict with other uses of $T$ (denoted the total number of tasks ) in the paper.
>
> ## (c) Non-standard or unclear formulations
>
> Thank you for the constructive comment. We agree that the original expression $(\theta_0 + \tau)X^\top$ was ambiguous for deep networks, since $\theta$ represents the entire parameter set rather than a single linear operator.
>
> To address this, we revised the formulation to a **layer-wise definition**, where $\theta^{(l)} _ 0, \tau^{(l)} _ i$  and $x^{(l)}$ denote the pretrained weight, task vector, and input activations of layer $l$.
> The losses are now expressed as:
> $$
> L^{(l)} _ {trans}( \tilde{\tau}^{(l)} _ t) = E _ {x^{(l)} \sim D _ {i \le t-1}} \Big[\ell\big((\theta _ {t-1}^{merged, (l)}+\tilde{\tau}^{(l)} _ t)\ x^{(l)}, \theta _ {t-1}^{merged, (l)} \ x^{(l)} \big) \Big], \quad\quad\quad(Eq. 3)
> $$
> and
> $$
> L^{(l)} _ {fid}( \tilde{\tau}^{(l)} _ t) = E _ {x^{(l)} \sim D _ t} \Big[\ell\big((\theta _ {t-1}^{merged, (l)}+\tilde{\tau}^{(l)} _ t)\ x^{(l)}, (\theta _ {0}^{(l)} + \tau^{(l)} _ t)\ x^{(l)} \big) \Big], \quad\quad\quad(Eq. 3)
> $$
>
> ## (d) Undefined acronyms (e.g., ECDF).
>
> Thank you for pointing this out. We have updated the manuscript to define the term upon first use as *empirical cumulative distribution functions (ECDFs)*. (L228)
>
> # Q1 — Hyper-parameter Selection
>
> Thank you for the thoughtful question. Similar to other data-free merging methods such as TA, TIES-Merging, and OPCM, the rank hyperparameters ($r _ p, r _ l, r _ v$) in our method are tuned using a small validation set. Although the framework is data-free during merging, hyperparameter selection follows the standard practice of using validation data from the target domain. In practice, we found that the default settings provided in the paper work reliably across backbones and task sequences, requiring minimal tuning.
>
> We did not employ data-free proxy heuristics (e.g., NTK curvature, spectrum-based criteria, or stability proxies) for hyperparameter selection. While such techniques are promising, designing fully data-independent rank selection remains challenging and is an interesting direction for future work.
>
> Empirically, the hyperparameters exhibit unimodal sensitivity and can be adjusted independently:
>
> - **$r _ p$** controls how much of the previous-task subspace is preserved; larger values provide stronger forgetting mitigation.
> - **$r _ l$** is set smaller than $r _ p$ to maintain the structure of the null space while allowing controlled flexibility.
> - **$r _ v$** governs the dimensionality of the task-vector subspace and is kept small for stable alignment.
>
> For completeness, we have added a short hyperparameter adjustment guide in the appendix.

---

> ### Author Response · Authors · 2025-11-24
> **Respond to Reviewer SpsW (3/3)**
>
> # Q2 — Selection of $r_v$ and the rank assumption
>
> Thank you for the question. We clarify here how the rank assumption in the theorem relates to the practical choice of a small $r_v $, and what implications this has.
>
> In the theorem, the task vector at layer $ l $ is decomposed as $\tau^{(l)} = T _ 0 + E,\quad T _ 0 = B H, rank(T _ 0) = r _ d$, where $T _ 0$ captures the structured, data-aligned component and $E$ is the residual. The quantity $r_d$ denotes the rank of $T _ 0$, i.e., the number of dominant directions along which the data representations can be well aligned. Importantly, $r_d$ is typically smaller than the full rank of the layer or the feature dimension.
>
> The theoretical assumption involves $ \sigma_{r _ d}(T _ 0) $ being sufficiently large, ensuring that these $ r _ d $ directions are well separated and informative. In practice, we set $ r_v $ to a **small value** to select only the leading singular directions of $ \tau^{(l)} $, which are most likely to correspond to those dominant directions in $ T _ 0 $. As $ r _ v $ increases, weaker singular vectors with much smaller singular values are included, which are less reliably aligned with the data and may violate the condition that $ \sigma _ {r _ d}(T _ 0) $ remains large.
>
> Thus, using a small $ r _ v $ is consistent with the spirit of the rank assumption: we intentionally restrict ourselves to the most significant directions where the alignment is theoretically justified. Empirically, we observe that small $ r _ v $ yields stable performance, which supports the use of a low-rank approximation for the data-aligned subspace in our framework.

---

### Official Review · Reviewer_FVQh · 2025-11-01

**Soundness:** 3
**Presentation:** 2
**Contribution:** 2
**Rating:** 4
**Confidence:** 4

**Summary:**

This paper presents NUFILT, a data-free continual model merging approaches with good empirical performance and theoretical grounding.

**Strengths:**

1. The writing and formatting are good and clear.
2. The authors provide detailed and intuitional theoretical derivations to arrive at the final solution.
3. NUFILT achieves good empirical performance on both vision and NLP tasks, outperforming previous train-free and train-required merging methods on this task.

**Weaknesses:**

1. Potential unfair baseline comparison: The baseline train-free merging methods (e.g. TA, TIES) are implemented with theta_merged_t = theta_merged_t-1 + lambda tau_t. This implementation biases the model towards the early task vectors. For example, the optimal lambda for the first task is clearly 1.0, while later tasks have gradually reducing optimal lambda values. This probably explains the result of task arithmetic (Table 1) having lower performance than even weight averaging. I think the more fair comparison to these approaches should include a downscaling of previously merged task vectors (similar to the implementation of weight averaging), that is: theta_merged_t = (t-1)/t * theta_merged_t-1 + (1/t) * theta_0 + (lambda / t) * tau_t.

2. Comparison to OPCM. The first step of NUFILT resembles OPCM, i.e., they both try to project the current task vector to the orthogonal space of previous merged vector. Given the similarity on the intuition, a more detailed comparison should be added to explicitly demonstrate the differences and novelties, especially on how it addresses the criticism to OPCM that it struggles when task vectors are inherently entangled or non-orthogonal (section 3.2).

3. Introduced new terminologies: I do not agree it is necessary or worth to introduce the new terms transparency and fidelity, when they are just new names for the classical Stability-Plasticity trade-off in continual learning. They both measure the adaptiveness to new task and forgetting on previous tasks. Furthermore, the first contribution in the introduction that “formulating these two desiderata frames a new open challenge absent from prior work” seems an over-claim.

4. Computational cost to train-free methods: Table 4 only compares the computational cost to the training-required baselines. It is necessary to compare to the training-free baselines (which have way lower computational cost) as well to demonstrate the performance-computation trade-off.

5. Scalability to larger models: The paper demonstrates that it can scale to ViT and T5 models, but did not provide results with larger models (e.g., Llama), is it possible to scale to even larger language models given the training-required procedure?

**Questions:**

1. Using LoRA. One key benefit of LoRA is that it updates only a lightweight set of parameters. In this case, the LoRA parameters are going to be incorporated to the base parameters, thus this advantage is lost. Is it necessary to use LoRA then, or maybe it still brings some training regularization?
2. The empirical results demonstrate good accuracy but relatively poorer backward transfer (Table 1). Have you tried using a weighted sum for the two objectives in Equation 16 to enforce less forgetting?
3. From the description of OPCM [1], the correct citation of the vision benchmarks should be [2] and [3].

[1] Merging Models on the Fly Without Retraining: A Sequential Approach to Scalable Continual Model Merging

[2] Editing Models with Task Arithmetic

[3] Localizing Task Information for Improved Model Merging and Compression

---

> ### Author Response · Authors · 2025-11-24
> **Respond to Reviewer FVQh (1/n)**
>
> # W1 — Potential unfair baseline comparison
>
> Thank you for raising this point. We clarify that the standard train-free baselines (e.g., TA and TIES) are implemented following their original update rule:$\theta^{(t)} _ {merged}= \theta^{(t-1)} _ {merged} + \lambda \tau _ t$. Expanding this recurrence gives $\theta^{(T)} _ {merged}= \theta _ 0 + \lambda \sum _ {t=1}^T \tau _ t$, showing that **each task vector contributes equally**, and the formulation does **not** introduce any bias toward early tasks. Thus, the implementation does not favor earlier tasks over later ones.
>
> The variant you suggested: $\theta^{(t)} _ {merged} = \frac{t-1}{t} \theta^{(t-1)} _ {merged} + \frac{1}{t} \theta _ 0 + \frac{\lambda}{t} \tau _ t$,  expands to $\theta^{(T)} _ {merged} = \theta _ 0 + \frac{\lambda}{T} \sum_{t=1}^T \tau _ t$, which simply rescales all task vectors by $1/T$. This normalization does not change the relative contributions of tasks, but it does reduce the overall update magnitude.
>
> For completeness, we evaluated this variant alongside the standard TA implementation. As shown in Table F.1, the variant performs worse than the original TA formulation, while our method substantially outperforms both. We also note that NUFILT is scale-free (i.e., it does not introduce a coefficient such as $\lambda$), suggesting that the comparison is not affected by task-vector scaling and that the baseline implementations are aligned with their intended formulations.
>
> Table F.1: Comparison of TA and TA (variant) on 8 tasks using CLIP ViT-B/32
>
> | Method         | 8 tasks   | 14 tasks  | 20 tasks  |
> | -------------- | --------- | --------- | --------- |
> | TA             | 67.5±0.0  | 66.5 ±0.0 | 60.0 ±0.0 |
> | TA (variant)   | 56.7±0.0  | 58.5±0.0  | 56.8±0.0  |
> | Weight Average | 66.3 ±0.0 | 65.4 ±0.0 | 61.1 ±0.0 |
> | NUFILT         | 83.6±0.2  | 78.0±0.2  | 71.0±0.9  |
>
> # W2 — Comparison to OPCM
>
> Thank you for the insightful comment. While NUFILT and OPCM both begin by projecting the incoming task vector against information from previous tasks, the two methods differ substantially in what subspace is used and how adaptation is handled.
>
> **OPCM** enforces *parameter-level orthogonality* by projecting the new task vector onto the null space of previously merged task vectors: $F(\tau _ t) = \sum _ {\substack{p,q=a , p\neq q}}^{m,n} \langle \tau _ t, u _ p^{(t-1)} v _ q^{(t-1)\top} \rangle_F u_p^{(t-1)} v _ q^{(t-1)\top}$, where the projection is constructed from both left and right singular vectors of $\tilde{\tau}_{\leq t-1}$. This strict orthogonality constraint may limit adaptation when task vectors are entangled or not well orthogonal.
>
> **NUFILT**, in contrast, constructs a projection using only the **right singular vectors** of the previously merged vectors: $P _ t^{(l)} = I - \hat{V} _ {\leq t-1}^{(l)} \hat{V} _ {\leq t-1}^{(l)\top}$, which approximates the **null space of the input representation space** rather than the parameter space.  After projection, NUFILT introduces a lightweight LoRA adaptation term: $F(\tau _ t) = \tau _ t^{(l)} \left( P _ t^{(l)} + B _ t^{(l)} A_t^{(l)} \right)$, which provides additional flexibility to adjust the update when the raw projection is insufficient. When task vectors are inherently entangled, strict orthogonality (as in OPCM) removes useful components, whereas NUFILT preserves them through the learnable low-rank correction.
>
> Results in Table 4 confirm this effect: in the 8-task ViT-B/32 setting, the null-space filtering component alone already outperforms OPCM (80.0 vs. 75.5), and adding the LoRA refinement further improves performance to 83.6. This illustrates the increased robustness of our method when task vectors are non-orthogonal or exhibit substantial overlap.

---

> ### Author Response · Authors · 2025-11-24
> **Respond to Reviewer FVQh (2/n)**
>
> # W3 — Introduced new terminologies
>
> Thank you for the comment. Stability and plasticity are indeed broad and widely used notions in continual learning. In our work, *transparency* and *fidelity* refer to **their concrete instantiations in the setting of continual model merging**, where the basic operations involve combining task vectors rather than updating models through data.
>
> Specifically, continual model merging requires reasoning about how task vectors interact in parameter space. Transparency and fidelity therefore capture the two key aspects of this interaction:
>
> -  **Transparency** formalizes how the new task vector affects the parameter responses associated with previously merged tasks.
> -  **Fidelity** formalizes how well the merged parameters preserve the contribution of the new task vector.
>
> These notions are thus aligned with classical continual learning goals, but expressed at the **parameter level** that is unique to model merging. We have updated the text to clarify this relationship and avoid potential ambiguity.
>
> We have updated Section 3.3 to more clearly articulate how these concepts relate to classical stability-plasticity concepts, and **removed** the statement about transparency and fidelity being main contributions in the introduction.
>
> # W4 — Computational cost to train-free methods
>
> Thank you for the suggestion. We have added a comparison against the training-free baselines (TA and OPCM) to clarify the performance–computation trade-off. As shown in Table 5, training-free methods are indeed the most efficient since they require no data loading or optimization, but their accuracy is notably lower. Data-dependent baselines such as LW AdaMerging and LoRA-WEMOE exhibit the highest total runtime due to multi-step training and data loading. Compared with WUDI-Merging, our method achieves lower solving time because its LoRA-based formulation optimizes far fewer parameters, while the total time is higher due to an additional SVD step. Overall, our method provides the strongest accuracy while maintaining competitive efficiency among data-free approaches.
>
> Table F.2: Overhead of ViT-B/32 under 8-task continual merging: per-task iterations, solving time, total time, peak GPU memory, and final parameter count
>
> | Model             | Iters. | Solv. | Tot.   | GPU Mem. | Param. | ACC (%)    |
> | ----------------- | ------ | ----- | ------ | -------- | ------ | ---------- |
> | **TA**            | –      | –     | 8.3s   | –        | 87.5M  | 67.5 ± 0.0 |
> | **OPCM**          | –      | –     | 76.6s  | 1.3GB    | 87.5M  | 75.5 ± 0.5 |
> | **LW AdaMerging** | 50     | 47.1s | 724.2s | 1.6GB    | 87.5M  | 53.4 ± 3.2 |
> | **LoRA-WEMOE**    | 50     | 72.9s | 754.6s | 1.8GB    | 103.7M | 68.8 ± 7.8 |
> | **WUDI-Merging**  | 50     | 28.9s | 37.7s  | 2.0GB    | 87.5M  | 74.7 ± 6.6 |
> | **NUFILT (ours)** | 25     | 9.3s  | 129.8s | 1.8GB    | 87.5M  | 83.3 ± 0.3 |
> | **NUFILT (ours)** | 100    | 35.9s | 156.4s | 1.8GB    | 87.5M  | 83.6 ± 0.2 |
> | **NUFILT (ours)** | 50     | 18.4s | 138.9s | 1.8GB    | 87.5M  | 83.6 ± 0.2 |
>
> # W5 — Scalability to larger models
>
> Thank you for raising this point. To address the scalability of our approach to larger models, we have extended our evaluation to include **LLaVA-1.5-7B**.
>
> Our experiments demonstrate that the proposed framework is capable of efficiently scaling to larger models, confirming the robustness and generalizability of our approach. Notably, our method is model-agnostic. We have previously validated it on CLIP (contrastive architecture) and Flan-T5 (encoder-decoder architecture), and now, we have extended it to LLaVA-1.5 (decoder-only architecture), further showcasing its versatility. We have included these results in the revised version of the paper. (see section 5.2)
>
> Table F.3: Results of continual merging LLaVA models on 4 tasks, ordered alphabetically.
>
> | Method       | IconQA | Image | SciQA | VizWiz | Average |
> | ------------ | ------ | ----- | ----- | ------ | ------- |
> | Individual   | 75.5   | 96.0  | 83.7  | 64.8   | 80.0    |
> | Zero-Shot    | 14.1   | 40.9  | 61.7  | 41.0   | 39.4    |
> | TA           | 47.6   | 67.9  | 74.2  | 46.6   | 59.1    |
> | TIES-MERGING | 69.6   | 74.4  | 75.5  | 44.6   | 66.0    |
> | OPCM         | 58.0   | 86.4  | 81.1  | 44.9   | 67.6    |
> | NUFILT       | 59.6   | 93.6  | 80.6  | 48.3   | 70.5    |

---

> ### Author Response · Authors · 2025-11-24
> **Respond to Reviewer FVQh (3/3)**
>
> # Q1 — Using LoRA
>
> Thank you for your question. We use LoRA specifically to adjust the projector  $P$ efficiently. The key benefits of using LoRA in our approach are: (1) **Efficiency**: LoRA introduces only a small set of additional parameters, making the fine-tuning process computationally efficient. (2) **Minimal Disruption**: LoRA allows us to make small adjustments to the projector  $P$ , thereby increasing the model’s fidelity without significantly distrupt transparency.
>
> Regarding the concern that "LoRA parameters are incorporated into the base parameters," we would like to clarify that LoRA is used to fine-tune the projector, and after merging, the number of parameters **remains the same as the base model.** Without LoRA merging, the model would require additional parameters for LoRA itself, which would make the model less parameter-efficient.
>
> # Q2 — Weighted Sum for the Two Objectives
>
> Thank you for your suggestion. We have not previously tuned the weights for the two objectives in Equation 16, as we aim to avoid introducing more hyperparameters. Following your suggestion, we experimented with weight adjustments for both objectives (using $w_1$ for the transparency term and $w_2$ for the fidelity term). The results show that varying the weights does not significantly change performance, demonstrating the robustness of our two optimization objectives with respect to the coefficient adjustments.
>
> Table F.4: Comparison of different weight combinations for the two objectives on 8 tasks using CLIP ViT-B/32
>
> | $w_1$ | $w_2$ | Accuracy | Backward Transfer |
> | ----- | ----- | -------- | ----------------- |
> | 1     | 1     | 83.6±0.2 | -2.7±0.7          |
> | 0.1   | 1     | 83.6±0.2 | -2.8±0.8          |
> | 1     | 0.1   | 83.4±0.3 | -2.7±0.7          |
> | 10    | 1     | 83.4±0.3 | -2.7±0.7          |
> | 1     | 10    | 83.6±0.3 | -2.9±0.8          |
>
> # Q3 —  Citation correction
>
> Thank you for your careful check! We have updated the citation to correctly reference [2] and [3] for the vision benchmarks.

---

> ### Comment · Reviewer_FVQh · 2025-11-25
>
> Thanks to the authors for their works in the rebuttal process. Most of my concerns have been addressed, such as comparison to prior works, scalability to larger models, additional ablation of weighted objectives. Given the clarifications and additional experiments, I have updated my final score to 6.
>
> Additionally, I have the following questions which do not affect my overall asessement.
> 1. Regarding the baseline implementation, I do not agree that the recurrence
>
> $\\theta^{\\{t\\}\\{\\text{merged}\\}} = \\theta^{\\{t-1\\}\\{\\text{merged}\\}} + \\lambda \\tau^{\\{t\\}}$
> can be expanded to
> $\\theta^{\\text{merged}} = \\theta_0 + \\lambda \\sum_{\\tau=1}^T \\tau^\\prime$
> because in the first equation (the continual merging case), the optimal $\\lambda$ is dependent on the previously merged model for each new task, while in the second equation (general model merging case) it assumes that each task here is independent. And therefore, I disagree that it shows "each task vector contributes equally" in this case. As demonstrated in my example in my earlier reply, I think the current way of implementation would lead to a decaying $\lambda$ value unless we downscale the merged task vector before merging new task vector. The reason why the downweighted version works even worse than the current version might be it needs a different tuning range of $\\lambda$, which should be larger than 1.0 (e.g., {1.0, 1.1, ...,}) while a constant $\\lambda=1.0$ would return the weight average baseline.
>
> 2. I still hold my view that it is more straightforward to use classical terms rather than introducing new terminologies which might confuse readers, as it has already confused 2 out of 4 reviewers in the reviewing process. I agree with the authors that in the continual model merging setting the merging should also account for parameter-level interferences (which are not expressed neither by the proposed new terms), but the overall goal here highly aligns with classical continual learning setting to absorb new tasks while preserving old task knowledge. For example, previous works in the continual model merging literature also chose to maintain the usage of classical terms Stability and Plasticity, while explaining them in the continual model merging setting [1].
>
> [1] Continual Model Merging without Data: Dual Projections for Balancing Stability and Plasticity; NeurIPS 2025

---

> ### Author Response · Authors · 2025-11-26
> **Respond to Reviewer FVQh**
>
> Thank you for taking the time to reevaluate our work and for raising these helpful points.
>
> (1) For your first point (baseline implementation of TA), we now realize our earlier explanation was not clear enough and may have caused a misunderstanding. In our experiments, we follow the TA implementation used in prior continual model merging works, in particular:
>
> [1] Continual Model Merging without Data: Dual Projections for Balancing Stability and Plasticity (NeurIPS 2025), and
>
> [2] Merging Models on the Fly Without Retraining: A Sequential Approach to Scalable Continual Model Merging (NeurIPS 2025).
>
> Our TA results are consistent with these references.
> Importantly, we do not choose a separate optimal λ for each incremental task. Instead, we use a single global λ shared across all T tasks (i.e., λ does not depend on t). The selection of λ is described in Appendix B.3. Under this setting, each task vector enters the merged model with the same scaling, so no task is favored simply because it appears earlier in the sequence.
>
> (2) For your second point (terminology), we agree with your concern that introducing new terms may not add much value and risks confusing readers. Following your suggestion, we have removed “transparency” and “fidelity” in the whole paper and now use the classical terms stability and plasticity throughout the paper. In Section 3.3, for example, we now describe these ideas as layer-wise surrogate quantities for stability and plasticity at the parameter level. The revised text (highlighted in blue in the updated PDF) is:
>
> >In continual model merging, updates for a new task should be incorporated without disrupting knowledge accumulated from earlier ones. This reflects the classical stability–plasticity balance in continual learning.
> >To describe how a transformed update $\tilde{\tau}^{(l)} _ t = F(\tau^{(l)} _ t)$ influences the merged model, we use two layer-wise quantities as surrogates for these effects. Let $\theta^{(l)} _ 0 \in \mathbb{R}^{d_o \times d _ i}$ be the pretrained weights at layer $l$,  $\tau^{(l)} _ i$ the task vector for task $i$,  and $x^{(l)}\in R^{d _ i}$ the corresponding layer activations.
>
> >Stability surrogate: This term measures how much the new update alters the model’s responses on earlier tasks.  For activations $x^{(l)} \sim D_{i\le t-1}$, we define: $L^{(l)}_{stab}( \tilde{\tau}^{(l)} _ t)=E _ {x^{(l)} \sim D _ {i\le t-1}}\big[\ell((\theta _ {t-1}^{merged,(l)}+\tilde{\tau}^{(l)} _ t)x^{(l)},\theta _ {t-1}^{merged,(l)}x^{(l)})\big]$,
>
> >Plasticity surrogate: This term evaluates how well the merged parameters follow the behavior of the task’s individual model.  For activations $x^{(l)} \sim D _ t $, we define: $L^{(l)} _ {plas}( \tilde{\tau}^{(l)} _ t)= E _ {x^{(l)} \sim D _ {t}}\big[\ell((\theta _ {t-1}^{merged,(l)}+\tilde{\tau}^{(l)} _ t)x^{(l)}, (\theta^{(l)} _ 0+\tau^{(l)} _ t)x^{(l)})\big]$.
>
> >These two terms anchor the stability–plasticity trade-off within the parameter space used for merging.
>
> We hope these clarifications address your concerns, and we would be glad to discuss further if any questions remain.

---

> > ### Comment · Reviewer_FVQh · 2025-11-26
> >
> > Thanks to the authors for their further clarifications and reply. My concerns were well addressed and I lean towards acceptance of their paper.

---

> > > ### Author Response · Authors · 2025-11-26
> > >
> > > We’re glad that we could address your concerns. Your feedback has clearly helped strengthen the paper. We truly appreciate it, thank you!

---

### Author Response · Authors · 2025-11-25
**Summary of Changes**

We thank all reviewers for their constructive feedback. Following the comments, we made the following revisions:

**Orthogonality & prior work — Reviewers m3Kf, FVQh and SpsW.**
 We expanded the related-work section to include a comparison with recent orthogonality-based methods and explained how NUFILT differs from OPCM in both the projection space and adaptation mechanism. We added comparisons with more recent SVD-based merging methods.

**Theory–practice alignment & hyperparameters — Reviewers m3Kf, GLxk, and SpsW.**
 We clarified how theoretical spectral quantities motivate the design without requiring data, unified the role of ($r _ p,r _ l,r _ v$),  and added concise guidance on their practical selection.

**Transparency, fidelity — Reviewers FVQh and SpsW.**
We explained the distinction between *transparency* and *fidelity* and their relation to classical stability–plasticity trade-offs in continual learning. We added quantitative evaluations for both objectives and removed the earlier claim of these as our main contributions.

**Efficiency & scalability — Reviewers m3Kf, GLxk, SpsW, and FVQh.**
 We incorporated full runtime results including SVD overhead and added new experiments on LLaVA-1.5-7B, confirming scalability to larger models.

We sincerely appreciate the reviewers' helpful comments and suggestions, which have significantly enhanced the clarity and rigor of the paper.

---

### Meta-Review · Area_Chair_GsyR · 2025-12-11

**Summary:**

The main weaknesses in the paper were mostly agreed upon; not enough comparisons or settings in which it works, too small, etc.; Writing issues; Termonology objections and a few specific ones.
In general, it seems that during rebuttal, most issues were addressed and reviewers already agreed to raise their scores, moreover, except the terminology (which, being a repeated concern, I suggest authors to think about whether they want to do more in that writing front) a lot of experiments were run to address issues.

**Reviewer Concerns:**

Terminology is maybe the only issue that is clearly big, and agreed upon issue. Many small issues were suggest and most of those were addressed.

**Reviewer Scores:**

That is not a fair, relevant or meaningful question. I protest the way this was all handled.
A Reviewers are not here, and ToM is weak, at least mine and the one literature study. I will not try to predict people.
B Scores are, anyway, a weak signal of interest; a paper should not be accepted or rejected just based on it. An AC's job is to look at the specific weaknesses and translate them into a recommendation.
C There are about 100 pages of discussions for me to read overall, in addition to the discussions I monitored and were just replaced, this is beyond my personal ability to do fairly. I did my best effort.

Saying all that, we know from their writing and practically, that this paper would have received high scores and strong support from reviewers.

---

### Decision · Program_Chairs · 2026-01-26

Accept (Poster)